# Nonlinearly Preconditioned Gradient Methods under Generalized Smoothness

**Konstantinos Oikonomidis** [1 2] **Jan Quan** [1 2] **Emanuel Laude** [3] **Panagiotis Patrinos** [1 2]

## Abstract

We analyze nonlinearly preconditioned gradient methods for solving smooth minimization problems. We introduce a generalized smoothness property, based on the notion of abstract convexity, that is broader than Lipschitz smoothness and provide sufficient first- and second-order conditions. Notably, our framework encapsulates algorithms associated with the gradient clipping method and brings out novel insights for the class of $(L_0, L_1)$-smooth functions that has received widespread interest recently, thus allowing us to extend beyond already established methods. We investigate the convergence of the proposed method in both the convex and nonconvex setting.

## 1. Introduction and preliminaries

We consider minimization problems of the form:

$$\min_{x \in \mathbb{R}^n} f(x), \tag{1}$$

where $f$ is a continuously differentiable and possibly nonconvex function. While gradient descent is a reliable solver for this type of problems, in many cases it does not fully take advantage of the cost function properties and requires well-tuned or costly stepsize strategies to converge.

In this paper we thus focus on nonlinearly preconditioned gradient methods that are tailored to the properties of the cost function. Given stepsizes $\gamma > 0$ and $\lambda > 0$ and a starting point $x^0 \in \mathbb{R}^n$, we consider the following iteration:

$$x^{k+1} = T_{\gamma,\lambda}(x^k) := x^k - \gamma \nabla \phi^*(\lambda \nabla f(x^k)), \tag{2}$$

where $\phi^* : \mathbb{R}^n \to \mathbb{R}$ is convex and is called the *dual reference function*. The convex conjugate of $\phi^*$, $\phi$ is called the *reference function* and its properties are crucial to our analysis. This general framework was originally analyzed in (Maddison et al., 2021) under the so-called dual relative smoothness condition for convex and essentially smooth functions. In the general nonconvex and composite nonsmooth setting, it was studied in (Laude & Patrinos, 2025) under a condition called the anisotropic descent property, which was itself first introduced in (Laude et al., 2023) and can be regarded as a globalization of anisotropic prox-regularity (Laude, 2021) for smooth functions.

Our analysis is mainly focused on two types of reference functions that are both generated by a kernel function $h : \mathbb{R} \to \mathbb{R}_+ \cup \{\infty\}$, thus resulting in two different families of algorithms. The first one is given by the composition with the Euclidean norm, i.e., $\phi = h \circ \| \cdot \|$ and is referred to as the *isotropic* reference function. In this case, the main iteration (2) takes the form of gradient descent with a scalar stepsize that depends on the iterates. The second is a separable sum obtained via $\phi(x) = \sum_{i=1}^{n} h(x_i)$, henceforth called the *anisotropic* reference function. In this case, (2) becomes gradient descent with a *coordinate-wise* stepsize that depends on the iterates of the algorithm. We remark that although the anisotropic reference function makes the analysis of the method more challenging, it generates more interesting algorithms, akin to the ones that are often used in practice.

### 1.1. Motivation

**Unifying framework for clipping algorithms.** Gradient clipping and signed gradient methods have garnered attention in recent years due to their efficiency in neural network training and other applications (Bernstein et al., 2018; Gorbunov et al., 2020; Zhang et al., 2020a;b;c; Koloskova et al., 2023; Kunstner et al., 2024). The intuition behind gradient clipping is straightforward, since by clipping one does not allow the potentially very large (stochastic) gradients to hinder the training. Nevertheless, in many cases the clipping threshold and stepsize should be carefully tuned in practice, otherwise leading to suboptimal performance (Koloskova et al., 2023). While algorithms of this type

---
[1]Department of Electrical Engineering (ESAT-STADIUS), KU Leuven, Kasteelpark Arenberg 10, 3001 Leuven, Belgium [2]Leuven.AI-KU Leuven Institute for AI, 3000 Leuven, Belgium [3]Proxima Fusion GmbH, Flößergasse 2, 81369 Munich, Germany. Correspondence to: Konstantinos Oikonomidis <konstantinos.oikonomidis@kuleuven.be>.

*Proceedings of the 42$^{nd}$ International Conference on Machine Learning*, Vancouver, Canada. PMLR 267, 2025. Copyright 2025 by the author(s).

*Table 1.* Examples of kernel functions along with the generated preconditioner and its (generalized) derivative in one dimension. The last column indicates whether Assumption 1.1 and Assumption 1.2 are satisfied, respectively.

| $h(x)$ | $\mathrm{dom}\, h$ | $h^{*\prime}(y)$ | $h^{*\prime\prime}(y)$ | Assumption 1.1/1.2 |
|---|---|---|---|---|
| $\cosh(x) - 1$ | $\mathbb{R}$ | $\mathrm{arcsinh}(y)$ | $\frac{1}{\sqrt{1+y^2}}$ | ✓/✓ |
| $\exp(\lvert x \rvert) - \lvert x \rvert - 1$ | $\mathbb{R}$ | $\ln(1 + \lvert y \rvert)\overline{\mathrm{sgn}}(y)$ | $\frac{1}{1+\lvert y \rvert}$ | ✓/✓ |
| $-\lvert x \rvert - \ln(1 - \lvert x \rvert)$ | $(-1, 1)$ | $\frac{y}{1+\lvert y \rvert}$ | $\frac{1}{(1+\lvert y \rvert)^2}$ | ✓/✓ |
| $1 - \sqrt{1 - x^2}$ | $[-1, 1]$ | $\frac{y}{\sqrt{1+y^2}}$ | $(1 + y^2)^{-3/2}$ | ✓/✓ |
| $x \,\mathrm{arctanh}(x) - \ln(\cosh(\mathrm{arctanh}(x)))$ | $(-1, 1)$ | $\tanh(y)$ | $1 - \tanh^2(y)$ | ✓/✓ |
| $\frac{1}{2}x^2 + \delta_{[-1,1]}(x)$ | $[-1, 1]$ | $\min(1, \max(-1, y))$ | $\partial_C(\Pi_{[-1,1]})(y)$ | ✓/✗ |

have been analyzed under various smoothness and stochasticity assumptions, there does not seem to exist a simple unifying framework that encapsulates them. Motivated by this gap, we propose a framework that provides further insights into existing methods but also naturally generates new algorithms.

**Majorization-minimization and $\Phi$-convexity.** A plethora of well-known optimization algorithms belong to the so-called majorization-minimization framework in that they are generated by successively minimizing upper bounds of the objective function. As a classical example, under Lipschitz smoothness of $f$, the celebrated gradient descent method with stepsize $1/L$ iteratively minimizes the following (global) quadratic upper bound around the current point $\bar{x} \in \mathbb{R}^n$:

$$f(x) \le f(\bar{x}) + \langle \nabla f(\bar{x}), x - \bar{x} \rangle + \frac{L}{2} \|x - \bar{x}\|^2.$$

It is straightforward that this inequality can be written as

$$f(x) \le f(\bar{x}) + \tfrac{1}{L}\phi(L(x - \bar{y})) - \tfrac{1}{L}\phi(L(\bar{x} - \bar{y})), \quad (3)$$

for $\phi = \frac{1}{2}\|\cdot\|^2$ and $\bar{y} = T_{L^{-1},1}(\bar{x}) = \bar{x} - \frac{1}{L}\nabla\phi^*(\nabla f(\bar{x}))$. Going beyond the standard Lipschitzian assumptions, it is natural to consider reference functions $\phi$ that generate less restrictive descent inequalities thus allowing us to efficiently tackle more general problems. This is exactly the anisotropic descent property (Laude & Patrinos, 2025, Definition 3.1) and minimizing this upper bound leads to the algorithm described in (2) (for $\lambda$ fixed). Considering (3), it is natural to examine reference functions that are strongly convex and grow faster than a quadratic in order to obtain a less restrictive descent inequality. It turns out that in many interesting cases, the preconditioner in (2) becomes similar to a sigmoid function and the algorithmic step takes the form of popular algorithms. We present examples of kernel functions and the corresponding preconditioners in Table 1.

The abstraction discussed in the previous paragraph is in fact tightly connected to the notion of $\Phi$-convexity (also known as $c$-concavity in the optimal transport literature),

which states that a function is $\Phi$-convex if it can be written as the pointwise supremum over a family of nonlinear functions. Similarly to the fact that every proper, lsc and convex function can be expressed as a pointwise supremum of its affine minorizers (Rockafellar & Wets, 1998, Theorem 8.13), anisotropic smoothness then requires that $-f$ is a pointwise supremum of nonlinear minorizers. This fact is once again in parallel to classical $L$-smoothness, which requires that $-f$ is the pointwise supremum over concave quadratics, and leads to an envelope representation of $f$ that is useful in studying the corresponding calculus. In that regard, anisotropic smoothness is a straightforward extension of Lipschitz smoothness. A visualization of the concept of $\Phi$-convexity is shown in Figure 1.

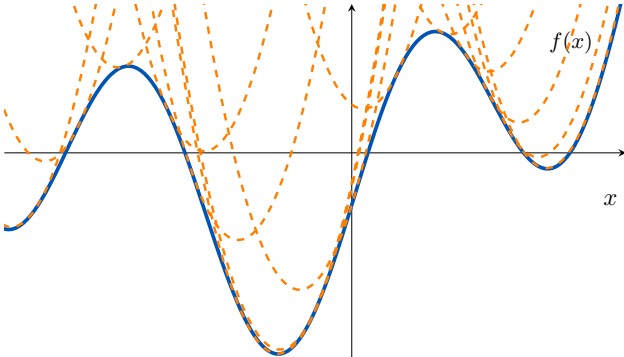

*Figure 1.* Visualization of the quadratic upper bounds of the function $f(x)$ at various points. By flipping the figure it can be seen that $-f$ is a $\Phi$-convex function: it is the pointwise supremum over concave quadratics of the form $-\phi(x - y) + \beta$, with $\phi = \frac{L}{2}\|\cdot\|^2$ and $y, \beta \in \mathbb{R}$. Note that this function is not convex in the classical sense, as there are no linear functions supporting it.

### 1.2. Our contribution

Our approach departs from and improves upon existing works in the following aspects.

- We describe a common nonlinear gradient precon-

ditioning scheme for the main iterates, i.e., without momentum nor exponential moving average mechanisms, of popular algorithms including gradient clipping, Adam, Adagrad and recently introduced methods for $(L_0, L_1)$-smoothness. These preconditioners are gradients of smooth convex functions and have sigmoid shape, reminiscent of common activation functions in neural networks.

- We introduce $(L, \bar{L})$-anisotropic smoothness, which extends (Laude & Patrinos, 2025) allowing for two constants and a reference function $\phi$ of possibly non-full domain and prove that it is less restrictive than $L$-smoothness. Through a novel technique we study necessary and sufficient first- and second-order conditions for $(L, \bar{L})$-anisotropic smoothness, in the process obtaining novel characterizations for the forward operator $T_{\gamma, \lambda}$, which in turn leads to new insights into the class of $(L_0, L_1)$-smooth functions.

- We analyze the convergence of (2) in the nonconvex setting, obtaining new results for our stationarity measure. In the convex setting, we prove the sublinear rate of the method for a large family of isotropic reference functions, utilizing only a simple dual characterization. In the more challenging case where $\phi$ is anisotropic, we present an unconventional proof that is based on the envelope representation of anisotropic smoothness. We are thus able to obtain standard $O(1/K)$ convergence rates for large classes of functions.

### 1.3. Related work

**Dual space preconditioning and anisotropic smoothness.** The scheme described in (2) was originally introduced in (Maddison et al., 2021) in the convex setting, where it was analyzed under a condition called dual relative smoothness for which sufficient second-order conditions were provided. In (Laude & Patrinos, 2025) the anisotropic smoothness condition was studied, which was shown to naturally lead to the convergence of the method in the nonconvex and proximal case. Moreover, in (Léger & Aubin-Frankowski, 2023) the scheme was also analyzed under the general framework of $\Phi$-convexity and a sufficient second-order condition for anisotropic smoothness was provided. Nevertheless, this requires that $\phi \in \mathcal{C}^4(\mathbb{R}^n)$ satisfies the non-negative cross curvature condition from optimal transport (NNCC) (see (Figalli et al., 2011, Assumption (B3)) and (Léger & Aubin-Frankowski, 2023, Definition 2.8)), which is a strong assumption that does not hold for many interesting reference functions. An accelerated version of the method for Lipschitz smooth problems was introduced and studied in (Kim et al., 2023). Recently, the method was also extended to measure spaces in (Bonet et al., 2024).

Furthermore, a relaxed proximal point algorithm with nonlinear preconditioning akin to (2) for solving monotone inclusion problems was studied in (Laude & Patrinos, 2023).

**Generalized smoothness.** Our work is also connected to other notions of generalized smoothness, i.e., descent inequalities beyond the standard Lipschitzian assumptions. To begin with, Bregman (relative) smoothness is a popular extension of Lipschitz smoothness (Bauschke et al., 2017a; Lu et al., 2018; Bolte et al., 2018; Ahookhosh et al., 2021) that can encapsulate a wide variety of functions such as those whose Hessians exhibit a certain polynomial growth (Lu et al., 2018). Other notions of generalized smoothness include Hölder smoothness (Bredies, 2008; Nesterov, 2015) and also higher-order smoothness where higher-order derivatives of $f$ are Lipschitz continuous (Nesterov & Polyak, 2006; Doikov & Nesterov, 2020).

Recently, a new concept of smoothness has been introduced in order to capture the cases where the norm of the Hessian is upper bounded by some function of the norm of the gradient (Zhang et al., 2020b; Chen et al., 2020; Li et al., 2024). This condition, which in its most popular form is called $(L_0, L_1)$-smoothness (Zhang et al., 2020b, Definition 1), has received widespread attention. Various existing methods have been analyzed under this new smoothness condition (Wang et al., 2023; Faw et al., 2023; Koloskova et al., 2023), while also new ones have been proposed (Gorbunov et al., 2024; Vankov et al., 2024). We remark that a number of these algorithms can actually be obtained in (2) via a suitable choice of the reference function. Nevertheless, it is important to note that in contrast to the aforementioned types of smoothness, anisotropic smoothness is not obtained via a linearization of the cost function around a point and thus it is not straightforward to compare the obtained descent lemmas.

### 1.4. Notation

We denote by $\langle \cdot, \cdot \rangle$ the standard Euclidean inner product on $\mathbb{R}^n$ and by $\| \cdot \|$ the standard Euclidean norm on $\mathbb{R}^n$ as well as the spectral norm for matrices. For a square matrix $A$ with real spectrum, $\lambda_{\max}(A)$ and $\lambda_{\min}(A)$ denote the largest and smallest eigenvalue respectively. We denote by $\mathcal{C}^k(Y)$ the class of functions which are $k$ times continuously differentiable on an open set $Y \subseteq \mathbb{R}^n$. For a proper function $f : \mathbb{R}^n \to \overline{\mathbb{R}}$ and $\lambda \geq 0$ we define the episcaling $(\lambda \star f)(x) = \lambda f(\lambda^{-1} x)$ for $\lambda > 0$ and $(\lambda \star f)(x) = \delta_{\{0\}}(x)$ otherwise. We adopt the notions of essential smoothness, essential strict convexity and Legendre functions from (Rockafellar, 1997, Section 26): we say that a proper, lsc and convex function $f : \mathbb{R}^n \to \overline{\mathbb{R}}$ is *essentially smooth* if $\mathrm{int}(\mathrm{dom}\, f) \neq \emptyset$ and $f$ is differentiable on $\mathrm{int}(\mathrm{dom}\, f)$ such that $\| \nabla f(x^\nu) \| \to \infty$, whenever $\mathrm{int}(\mathrm{dom}\, f) \ni x^\nu \to x \in \mathrm{bdry}\,\mathrm{dom}\, f$, and *essen-*

*tially strictly convex*, if $f$ is strictly convex on every convex subset of $\operatorname{dom} \partial f$, and *Legendre*, if $f$ is both essentially smooth and essentially strictly convex. In particular, a smooth convex function on $\mathbb{R}^n$ is essentially smooth.

Let $F : \mathbb{R}^n \to \mathbb{R}^n$ be a locally Lipschitz function, we denote the (Clarke) generalized Jacobian as $\partial_C F(x) = \operatorname{con}\{\lim_{x_i \to x} \nabla F(x_i) : x_i \notin \Omega_F\}$, where $\Omega_F$ is the set of points where $F$ fails to be differentiable. $\Pi_C$ denotes the projection on a set $C$. For an $f \in \mathcal{C}^2(\mathbb{R}^n)$ we say that it is $(L_0, L_1)$-smooth for some $L_0, L_1 > 0$ if it holds that $\|\nabla^2 f(x)\| \leq L_0 + L_1\|\nabla f(x)\|$ for all $x \in \mathbb{R}^n$. Otherwise we adopt the notation from (Rockafellar & Wets, 1998).

For clarity of exposition, for a vector $x \in \mathbb{R}^n$ we consider the function $\overline{\operatorname{sgn}}(x) = x/\|x\|$ for $x \in \mathbb{R}^n \setminus \{0\}$ and 0 otherwise.

### 1.5. Assumptions on the reference function

Our assumptions on the reference function $\phi$ are formulated as follows.

**Assumption 1.1.** *The reference function $\phi : \mathbb{R}^n \to \overline{\mathbb{R}}$ is proper, lsc, strongly convex and even, i.e., $\phi(x) = \phi(-x)$, with $\phi(0) = 0$.*

Assumption 1.1 is considered valid throughout the paper. Note that through the duality of strong convexity and Lipschitz smoothness, it implies that $\phi^* \in \mathcal{C}^1(\mathbb{R}^n)$. Moreover, from (Bauschke et al., 2017b, Proposition 11.7), $\{0\} = \arg\min \phi$ and thus $\phi \geq 0$. Throughout the paper we also consider specifically the case where $\phi^* \in \mathcal{C}^2(\mathbb{R}^n)$, for which we encode a sufficient condition in the following assumption, in light of (Rockafellar, 1977, p. 42).

**Assumption 1.2.** *$\phi \in \mathcal{C}^2(\operatorname{int} \operatorname{dom} \phi)$ is essentially smooth.*

It is important to note that through (Rockafellar, 1977, p. 42), under Assumptions 1.1 and 1.2, the Hessian matrix of $\phi^*$ is positive-definite everywhere.

Although we state both assumptions for the reference function $\phi$, we also use them throughout the paper by a slight abuse of notation for the kernel function $h$ which generates $\phi$.

When considering a kernel function $h$, in the anisotropic case, it is straightforward that $\phi$ inherits the properties of $h$ and the preconditioner takes the form $\nabla \phi^*(x) = (h^{*\prime}(x_1), \dots, h^{*\prime}(x_n))$. In the isotropic case, the differentiability of $\phi^*$ depends on the properties of $h$, as we show next.

**Lemma 1.3.** *Let $h : \mathbb{R} \to \overline{\mathbb{R}}$ satisfy Assumption 1.1. Then $h^* \geq 0$ is an even function and increasing on $\mathbb{R}_+$, while $h^{*\prime}(0) = 0$. Moreover, $\phi = h \circ \|\cdot\|$ is strongly convex, $\phi^* = h^* \circ \|\cdot\|$ and*

$$\nabla \phi^*(y) = h^{*\prime}(\|y\|)\overline{\operatorname{sgn}}(y), \qquad \forall y \in \mathbb{R}^n.$$

*If, furthermore, $h$ satisfies Assumption 1.2, then $\phi^* \in \mathcal{C}^2(\mathbb{R}^n)$.*

We provide examples of interesting kernel functions, along with the assumptions that they satisfy, in Table 1.

### 1.6. Connections with existing methods

As already mentioned, the scheme presented in (2) encompasses the basic iterations of various algorithms that are widely used in practice. In this subsection we thus provide some examples that showcase the generalizing properties of our framework.

*Example* 1.4. The standard gradient descent method can be obtained from (2) by choosing $\phi = \frac{1}{2}\|\cdot\|^2$.

*Example* 1.5. Let $\phi(x) = \sum_{i=1}^n 1 - \sqrt{1 - x_i^2}$. Then, (2) becomes

$$x_i^{k+1} = x_i^k - \gamma \frac{\nabla_i f(x^k)}{\sqrt{1/\lambda^2 + (\nabla_i f(x^k))^2}}$$

and by choosing $\lambda = \varepsilon^{-1/2}$ for some $\varepsilon > 0$ we retrieve the form of Adagrad (Duchi et al., 2011) without memory from (Défossez et al., 2022, Equation (3)) with $\beta_1 = \beta_2 = 0$.

*Example* 1.6. Let $\phi(x) = \sum_{i=1}^n -|x_i| - \ln(1 - |x_i|)$. Then, the main iterate (2) takes the following form:

$$x_i^{k+1} = x_i^k - \gamma \frac{\nabla_i f(x^k)}{1/\lambda + |\nabla_i f(x^k)|}.$$

and by choosing $\lambda = \varepsilon^{-1}$ for $\varepsilon > 0$, we retrieve the iterates of Adam (Kingma & Ba, 2014, Algorithm 1) where both the exponential decay rates are set to 0, i.e. $\beta_1 = \beta_2 = 0$.

*Example* 1.7. Let $h(x) = \frac{1}{2}x^2 + \delta_{[-1,1]}(x)$ and $\phi = h \circ \|\cdot\|$. Then, (2) becomes

$$x^{k+1} = x^k - \min(\gamma/\|\nabla f(x^k)\|, \gamma\lambda)\nabla f(x^k).$$

Note that the gradient clipping method as presented in (Zhang et al., 2020b, Equation (5)) is given by $x^{k+1} = x^k - \min\{\eta_c, \tilde{\gamma}\eta_c/\|\nabla f(x^k)\|\}\nabla f(x^k)$. Therefore, by choosing $\gamma = \tilde{\gamma}\eta_c$ and $\lambda = 1/\tilde{\gamma}$ we can see that (2) encompasses the gradient clipping method.

## 2. The extended anisotropic descent inequality

In this section we extend the definition of anisotropic smoothness from (Laude & Patrinos, 2025) to our setting where $\phi$ is potentially nonsmooth and provide sufficient conditions for a smooth function $f$ to satisfy this generalized descent inequality. The proofs can be found in Appendix B.

We begin with the definition of our extension of anisotropic smoothness.

**Definition 2.1** $((L, \bar{L})$-anisotropic smoothness**)**. *Let* $f \in \mathcal{C}^1(\mathbb{R}^n)$. *We say that* $f$ *is* $(L, \bar{L})$-*anisotropically smooth relative to* $\phi$ *with constants* $L, \bar{L} > 0$ *if for all* $x, \bar{x} \in \mathbb{R}^n$

$$f(x) \leq f(\bar{x}) + \bar{L} \left[ (L^{-1} \star \phi)(x - \bar{y}) \right. \\ \left. - (L^{-1} \star \phi)(\bar{x} - \bar{y}) \right], \quad (4)$$

*where* $\bar{y} = T_{L^{-1}, \bar{L}^{-1}}(\bar{x}) = \bar{x} - L^{-1} \nabla \phi^*(\bar{L}^{-1} \nabla f(\bar{x}))$.

Note that inequality (4) is well-defined for $\phi$ without full domain, but does not provide information for points $x \in \mathbb{R}^n$ such that $L(x - \bar{y}) \notin \operatorname{dom} \phi$. The intuition behind this extension of (Laude & Patrinos, 2025) is straightforward: we allow for two different smoothness constants that play a complementary role, while allowing $\operatorname{dom} \phi \neq \mathbb{R}^n$ leads to a more general descent inequality. It can be checked that for $\operatorname{dom} \phi = \mathbb{R}^n$, Definition 2.1 reduces to (Laude & Patrinos, 2025, Definition 3.1) but w.r.t. $\bar{L}\phi$.

Our first result is an extension of the envelope representation of $f$ under anisotropic smoothness (Laude & Patrinos, 2025, Proposition 4.1) to our setting where $\phi$ possibly does not have full domain.

**Proposition 2.2.** *Let* $f : \mathbb{R}^n \to \mathbb{R}$ *satisfy Definition 2.1. Then,* $f(x) = \inf_{y \in \mathbb{R}^n} \bar{L}(L^{-1} \star \phi)(x - y) + \xi(y)$ *for some proper* $\xi : \mathbb{R}^n \to \overline{\mathbb{R}}$. *Moreover,* $f^* = \psi + L^{-1}(\bar{L} \star \phi^*)$ *for some lsc and convex* $\psi : \mathbb{R}^n \to \overline{\mathbb{R}}$, *implying that* $f^* - L^{-1}(\bar{L} \star \phi^*)$ *is convex.*

Note that Proposition 2.2 describes one direction of a conjugate duality between anisotropic smoothness and Bregman (relative) strong convexity (Lu et al., 2018, Definition 1.2). This result along with the envelope representation of $f$ will be utilized later on in order to describe the convergence of the method in the convex setting.

In this paper we are mostly interested in cost functions that are not covered by the classical $L$-smoothness assumption and thus study reference functions $\phi$ that generate a less restrictive descent inequality. Therefore, we next show that for strongly convex reference functions, the class of anisotropically smooth functions is at least as large as that of Lipschitz smooth ones.

**Proposition 2.3.** *Let* $f : \mathbb{R}^n \to \mathbb{R}$ *be Lipschitz smooth with modulo* $L_f$ *and* $\phi$ *satisfy Assumption 1.1 with strong convexity parameter* $\mu$. *Then* $f$ *is* $(L_f/\mu, 1)$-*anisotropically smooth relative to* $\phi$.

### 2.1. Second-order sufficient conditions

In contrast to Euclidean or Bregman smoothness, anisotropic smoothness cannot in general be directly obtained via a second-order condition. This fact becomes apparent by noting that (4) is equivalent to $\bar{x} \in \arg\min_x g(x) := \bar{L}(L^{-1} \star \phi)(x - \bar{y}) - f(x)$. The second-order necessary condition for the minimality of $\bar{x}$ under As-

sumption 1.2 then becomes $\bar{L}L\nabla^2\phi(\nabla\phi^*(\bar{L}^{-1}\nabla f(\bar{x}))) - \nabla^2 f(\bar{x}) \succeq 0$, which does not normally imply the convexity of $g$. From the implicit function theorem, the above expression can be written as $\bar{L}L[\nabla^2\phi^*(\bar{L}^{-1}\nabla f(\bar{x}))]^{-1} - \nabla^2 f(\bar{x}) \succeq 0$, which is the form that we consider throughout the paper. This condition becomes sufficient when $f, \phi \in \mathcal{C}^2(\mathbb{R}^n)$ are Legendre through (Laude & Patrinos, 2025, Proposition 4.1). Harnessing the connection with $\Phi$-convexity, it is sufficient for general $f$ when $\phi$ is a regular optimal transport cost in light of (Villani, 2008, Theorem 12.46). Nevertheless, the regularity of $\phi$ is in general hard to verify, since its equivalent form requires the computation of fourth-order derivatives (Villani, 2008, Definition 12.27), and quite restrictive, not holding for many interesting functions. We thus follow a different strategy and study the minimization of $g$ using tools from optimization and nonsmooth analysis.

**Definition 2.4.** *Let* $f \in \mathcal{C}^2(\mathbb{R}^n)$. $f$ *satisfies the second-order characterization for* $(L, \bar{L})$-*anisotropic smoothness if for all* $x \in \mathbb{R}^n$ *and* $H \in \partial_C(\nabla\phi^*)(\bar{L}^{-1}\nabla f(x))$,

$$\lambda_{\max}(H\nabla^2 f(x)) < L\bar{L} \quad (5)$$

*and* $\lim_{\|x\| \to \infty} \|T_{L^{-1}, \bar{L}^{-1}}(x)\| = \infty$. *In particular,* (5) *reduces to* $\lambda_{\max}(\nabla^2\phi^*(\bar{L}^{-1}\nabla f(x))\nabla^2 f(x)) < L\bar{L}$ *under Assumption 1.2.*

Note that, since $\phi^*$ is Lipschitz smooth and convex, from (Hiriart-Urruty et al., 1984, Example 2.2) we know that $H$ is always a positive semi-definite matrix and thus from (Horn & Johnson, 2012, Theorem 1.3.22) with $A = H^{1/2}\nabla^2 f(x)$ and $B = H^{1/2}$ we have that $H\nabla^2 f(x)$ has real eigenvalues.

The generalized Jacobian considered in the definition above can in fact be computed for many interesting reference functions. As an example we study the inequality (5) for the reference function of Example 1.7 in Appendix D.2. The coercivity assumption on the forward operator in Definition 2.4 is very mild. For example consider a reference function $\phi$ with $\operatorname{dom} \phi \subseteq \overline{\mathbb{B}}(0, 1)$ as the isotropic ones generated by the kernels in the four last rows of Table 1. Then, by standard convex conjugacy, $\|\nabla\phi^*\| \leq 1$ and $\lim_{\|x\| \to \infty} \|T_{\gamma, \lambda}(x)\| = \infty$ always. We provide more results regarding the norm-coercivity property of $T_{L^{-1}, \bar{L}^{-1}}$ in Appendix D.1. It is important to note that when the matrix $H\nabla^2 f(x)$ is symmetric, we can remove this extra condition on the forward operator, as we show in Appendix D.

Under Assumption 1.2 we obtain a condition that is more straightforward to check.

**Lemma 2.5.** *Let* $\phi$ *satisfy Assumption 1.2 and* $f \in \mathcal{C}^2(\mathbb{R}^n)$. *Then,* (5) *holds if and only if for all* $x \in \mathbb{R}^n$

$$\nabla^2 f(x) \prec L\bar{L}[\nabla^2\phi^*(\bar{L}^{-1}\nabla f(x))]^{-1}. \quad (6)$$

*A sufficient condition for* (6) *is given by*

$$\lambda_{\max}(\nabla^2 f(x)) < L\bar{L}\lambda_{\min}([\nabla^2 \phi^*(\bar{L}^{-1}\nabla f(x))]^{-1}). \quad (7)$$

Next, we show that under Definition 2.4 the forward operator is actually a global homeomorphism (Facchinei & Pang, 2003, Definition 2.1.9), which we further utilize later on in proving our main result regarding the sufficiency of the second-order condition, Proposition 2.9.

**Proposition 2.6.** *Let* $f \in \mathcal{C}^2(\mathbb{R}^n)$ *satisfy Definition 2.4. Then,* $T_{L^{-1},\bar{L}^{-1}}$ *is injective.*

As a byproduct of our analysis, we obtain novel insights into the class of $\mathcal{C}^2(\mathbb{R}^n)$ $(L_0, L_1)$-smooth functions.

**Corollary 2.7.** *Let* $f \in \mathcal{C}^2(\mathbb{R}^n)$ *be* $(L_0, L_1)$-*smooth. Then,* $T_{\delta L^{-1},\bar{L}^{-1}} = x - \frac{\delta}{L_0 + L_1\|\nabla f(x)\|}\nabla f(x)$ *is monotone for* $\delta = 1$ *and strongly monotone for* $0 < \delta < 1$.

*Remark* 2.8. Considering the forward operator as defined in Corollary 2.7, the main iteration (2) becomes:

$$x^{k+1} = x^k - \frac{\delta}{L_0 + L_1\|\nabla f(x^k)\|}\nabla f(x^k). \quad (8)$$

Note that in this case, (Gorbunov et al., 2024, Algorithm 1) can be viewed as (2) with a conservative choice of $\delta \leq 0.57$. This algorithm is in fact generated by $\phi = h \circ \|\cdot\|$ with $h(x) = -|x| - \ln(1 - |x|)$.

Although Corollary 2.7 establishes new characterizations for $(L_0, L_1)$-smoothness, the simplification used in the proof of the second-order condition is quite restrictive. We provide examples where we can obtain tighter constants utilizing directly Definition 2.4 in Appendix D.

Having obtained a sufficient condition for the injectivity of the forward operator in Proposition 2.6, we now move on to providing sufficient conditions for $f$ to be $(L, \bar{L})$-anisotropically smooth.

**Proposition 2.9.** *Let* $f \in \mathcal{C}^1(\mathbb{R}^n)$ *and* $T_{L^{-1},\bar{L}^{-1}}$ *be injective. Let moreover either (i)* $\text{dom}\,\phi$ *be bounded or (ii)* $\text{dom}\,\phi = \mathbb{R}^n$ *and the following growth condition hold*

$$f(x) \leq \bar{L}(r^{-1} \star \phi)(x) - \beta$$

*for all* $x \in \mathbb{R}^n$ *and some* $r, \beta \in \mathbb{R}$ *such that* $0 < r < L$. *Then,* $f$ *is* $(L, \bar{L})$-*anisotropically smooth relative to* $\phi$.

*Remark* 2.10. The growth condition assumed in Proposition 2.9 when $\text{dom}\,\phi = \mathbb{R}^n$ is in fact not restrictive. Consider $\phi = \cosh \circ \|\cdot\| - 1$ and let $f$ be bounded above by some polynomial of the norm. Then, $\lim_{\|x\| \to \infty} \bar{L}(r^{-1} \star \phi)(x) - f(x) = +\infty$ for any fixed constants $\bar{L}, r$, implying that $\bar{L}(r^{-1} \star \phi)(x) - f(x)$ is lower bounded.

By combining Corollary 2.7 and Proposition 2.9 we can easily obtain the following result that describes the relation between $(L, \bar{L})$-anisotropic smoothness and $(L_0, L_1)$-smoothness.

**Corollary 2.11.** *Let* $f \in \mathcal{C}^2(\mathbb{R}^n)$ *be* $(L_0, L_1)$-*smooth. Then,* $f$ *is* $(\delta L_1, L_0/L_1)$-*anisotropically smooth relative to* $\phi(x) = -\|x\| - \ln(1 - \|x\|)$ *with* $\delta \in (0, 1)$.

### 2.2. How to compute the second-order condition

In this subsection we demonstrate how the second-order condition of Definition 2.4 can be computed for the two different instances of our preconditioned scheme. We consider kernel functions that satisfy Assumptions 1.1 and 1.2 implying that the results of Lemma 1.3 hold, lifting thus the need to compute generalized Jacobians.

**Anisotropic reference functions.** Thanks to the separability of $\phi$, the condition is simple to compute, since $[\nabla^2 \phi^*(\bar{L}^{-1}\nabla f(x))]^{-1}$ is just a diagonal matrix with elements $\alpha_{ii}$ given by

$$\alpha_{ii} = 1/h^{*''}(\bar{L}^{-1}\nabla_i f(x)) \quad (9)$$

**Isotropic reference functions.** As already mentioned in the introduction, in the isotropic case $\phi = h \circ \|\cdot\|$ the differentiability properties of $\phi^*$ depend on those of $h^*$. In the setting considered in this subsection though, we have $\nabla\phi^*(y) = h^{*'}(\|y\|)\overline{\text{sgn}}(y)$ with $\phi^* \in \mathcal{C}^2(\mathbb{R}^n)$ and the Hessian is given by

$$[\nabla^2 \phi^*(y)]^{-1} = \frac{1}{h^{*''}(\|y\|)}\frac{yy^\top}{\|y\|^2} + \frac{\|y\|}{h^{*'}(\|y\|)}\left(I - \frac{yy^\top}{\|y\|^2}\right)$$

for $y \in \mathbb{R}^n \setminus \{0\}$ and $[\nabla^2 \phi^*(y)]^{-1} = 1/h^{*''}(\|y\|)I$ otherwise. We provide examples of this condition for kernel functions displayed in Table 1 in Appendix D.

## 3. Algorithmic analysis

In this section we study the convergence properties of the method in the nonconvex and convex setting. The following assumption is considered valid throughout the remainder of the paper. The proofs of this section are deferred to Appendix C.

**Assumption 3.1.** $f \in \mathcal{C}^1(\mathbb{R}^n)$ *is* $(L, \bar{L})$-*anisotropically smooth relative to* $\phi$ *and* $f^\star = \inf f > -\infty$.

### 3.1. Nonconvex setting

The convergence of the method to stationary points, in the composite case with an additional nonconvex, nonsmooth term, was established in (Laude & Patrinos, 2025, Theorem 5.3) with a fixed stepsize $\gamma \leq L^{-1}$ and in (Laude & Patrinos, 2025, Theorem 5.5) using an adaptive linesearch strategy for choosing $\gamma$, always under the assumption that $\phi$ is of full domain. In the smooth setting studied in this paper, where $\phi$ is also even, we can improve upon the aforementioned results and show that stepsizes $\gamma$ up to $2L^{-1}$ can actually be used in the algorithm.

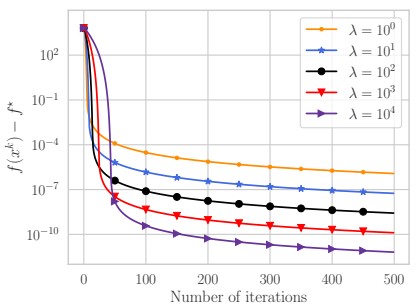 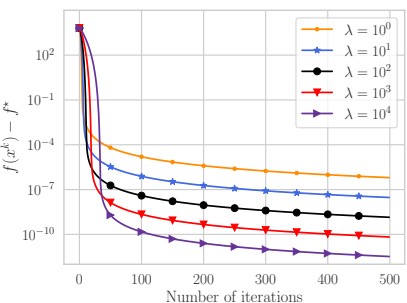 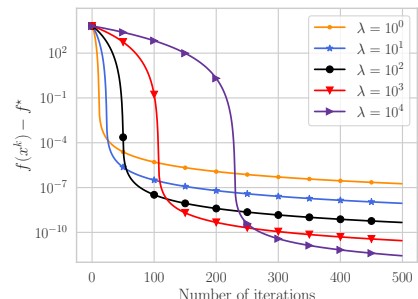

*Figure 2.* Minimizing $\frac{1}{4}\|x\|^4$ using (2). The figure on the left corresponds to $\phi_1(x) = \cosh(\|x\|) - 1$, the middle one to $\phi_2(x) = \exp(\|x\|) - \|x\| - 1$ and the one on the right to $\phi_3(x) = -\|x\| - \ln(1 - \|x\|)$. We choose values of $\bar{L}$, set $\lambda = \bar{L}^{-1}$ and then compute $\gamma = L^{-1}$ with $L$ as in Appendix D.

**Theorem 3.2.** *Let Assumption 3.1 hold and $\{x^k\}_{k\in\mathbb{N}_0}$ be the sequence of iterates generated by (2) with $\gamma = \alpha L^{-1}$, $\alpha \in (0,2)$, $\lambda = \bar{L}^{-1}$ and let $\beta = 1 - |1 - \alpha|$. Then we have the following rate:*

$$\min_{0 \le k \le K} \phi(\nabla\phi^*(\bar{L}^{-1}\nabla f(x^k))) \le \frac{L(f(x^0) - f^\star)}{\bar{L}\beta(K+1)}.$$

Using the result of Theorem 3.2 and specifying the reference function $\phi$, we can obtain convergence guarantees for the standard stationarity measure, $\|\nabla f(x^k)\|$. For $\phi = \cosh \circ \|\cdot\| - 1$, this is captured in the following corollary.

**Corollary 3.3.** *Let Assumption 3.1 hold, $\phi = \cosh \circ \|\cdot\| - 1$ and $\{x^k\}_{k\in\mathbb{N}_0}$ be the sequence of iterates generated by (2) with $\gamma = \alpha L^{-1}$, $\alpha \in (0,2)$, $\lambda = \bar{L}^{-1}$ and $\beta = 1 - |1 - \alpha|$. Then, the following holds for $P_0 = f(x^0) - f^\star$:*

$$\min_{0 \le k \le K} \|\nabla f(x^k)\| \le \sqrt{\frac{2L\bar{L}P_0}{\beta(K+1)}} + \frac{LP_0}{\beta(K+1)}.$$

### 3.2. Convex setting

In the convex setting, Proposition 2.2 establishes a useful connection between anisotropic smoothness and the convexity of $f^* - L^{-1}(\bar{L} \star \phi^*)$, i.e., the strong convexity of $f^*$ relative to $\bar{L} \star \phi^*$ with constant $L^{-1}$ in the Bregman sense. We utilize this connection in Proposition 3.5 and Theorem 3.6 in order to obtain standard sublinear convergence rates for the suboptimality gap in the isotropic case.

Henceforth we make the following standard assumption, which we consider valid throughout the rest of the paper unless stated otherwise.

**Assumption 3.4.** $\arg\min f \ne \emptyset$.

Our first result is the following novel characterization regarding the minimizers of $f$.

**Proposition 3.5.** *Let $x \in \mathbb{R}^n$ and $x^\star \in \arg\min f$. Moreover, let $f$ be $(L, \bar{L})$-anisotropically smooth relative to $\phi$*

*and convex. Then, the following inequality holds:*

$$\langle \nabla f(x), x - x^\star \rangle \ge L^{-1}\langle \nabla\phi^*(\bar{L}^{-1}\nabla f(x)), \nabla f(x)\rangle.$$

Obtaining sublinear rates for the function values is not a straightforward task. To the best of our knowledge, there do not exist such guarantees for the full generality of the setting we consider in this paper. Proposition 3.5 is useful in that regard, since it allows us to show a $O(1/K)$ rate for the suboptimality gap in the isotropic case, as we show next.

**Theorem 3.6.** *Let Assumption 3.1 hold, $f$ be convex and $\phi = h \circ \|\cdot\|$ with $h$ satisfying Assumption 1.1. For $\{x^k\}_{k\in\mathbb{N}_0}$ the sequence of iterates generated by (2) with $\gamma = L^{-1}$ and $\lambda = \bar{L}^{-1}$, the following holds:*

$$\|x^{k+1} - x^\star\| \le \|x^k - x^\star\|, \qquad (10)$$

*where $x^\star \in \arg\min f$, i.e. $\{x^k\}_{k\in\mathbb{N}_0}$ is Fejér monotone w.r.t. $\arg\min f$. Moreover, the norm of the gradient of $f$ monotonically decreases along the iterates of the algorithm:*

$$\|\nabla f(x^{k+1})\| \le \|\nabla f(x^k)\|,$$

*for all $k \in \mathbb{N}_0$. If, in addition, $h^{*\prime}(x)/x$ is a decreasing function on $\mathbb{R}_+$, we have the following rate for the suboptimality gap:*

$$f(x^K) - f^\star \le \frac{L\|\nabla f(x^0)\|\|x^0 - x^\star\|^2}{h^{*\prime}(\|\bar{L}^{-1}\nabla f(x^0)\|)(K+1)} \qquad (11)$$

We remark that $h^{*\prime}(x)/x$ being a decreasing function on $\mathbb{R}_+$ is in fact a mild assumption that holds for all the kernel functions presented in Table 1. The result above strengthens the known results regarding the convergence of the method from (Maddison et al., 2021) and (Laude & Patrinos, 2025), while also answering the question posed in (Maddison et al., 2021, p. 17), regarding obtaining convergence guarantees for the suboptimality gap $f(x^k) - f^\star$. We remark that although the obtained rate in (11) depends on

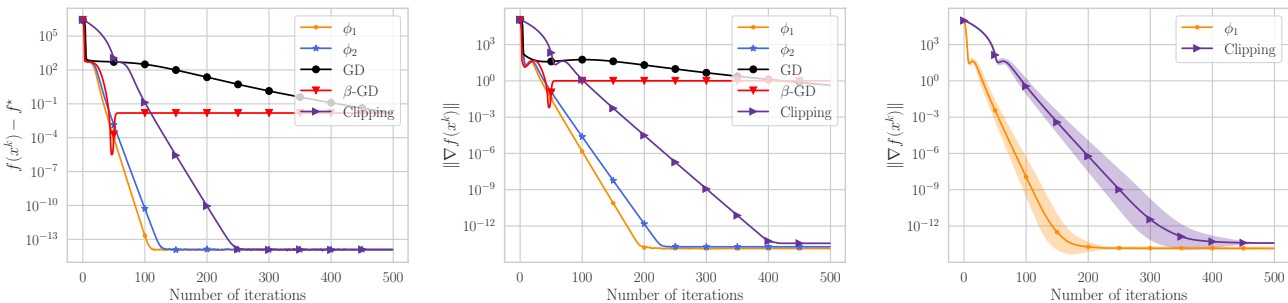

*Figure 3.* Nonconvex phase retrieval. $\phi_1$ corresponds to the isotropic reference function and $\phi_2$ to the anisotropic one, both of which are generated by $\cosh(\cdot) - 1$. The two figures on the left compare the algorithms for one instance of the problem. The figure on the right displays the results of gradient clipping and the isotropic version of (2) averaged across 100 random instances.

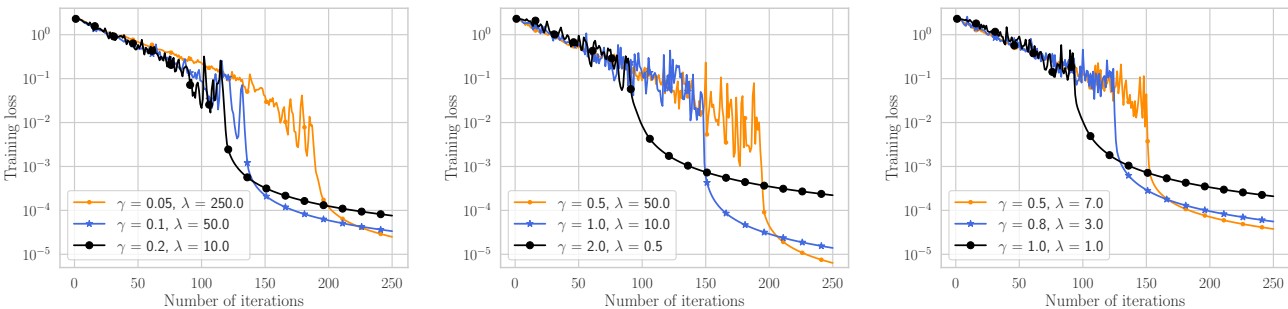

*Figure 4.* Simple NN training. (left) results for (2) with $\phi_1(x) = \cosh(\|x\|) - 1$; (middle) $\phi_2(x) = -\|x\| - \ln(1 - \|x\|)$; (right) gradient clipping method as presented in Example 1.7.

the initial norm of the gradient, one can use the techniques from (Vankov et al., 2024) or (Gorbunov et al., 2024) to achieve a better complexity when specifying the reference function $\phi$. Nevertheless, such an endeavor is beyond the scope of this paper.

Although Theorem 3.6 provides a sublinear rate for the isotropic case, obtaining such guarantees for anisotropic reference functions is not straightforward: key in the proof of Theorem 3.6 is the fact that $\nabla\phi^*(\bar{L}^{-1}\nabla f(x^k)) = h^{*\prime}(\|\bar{L}^{-1}\nabla f(x^k)\|)\overline{\mathrm{sgn}}(\nabla f(x^k))$ and therefore the convex gradient inequality for $f$ can directly be utilized to show the Fejér monotonocity of $\{x^k\}_{k\in\mathbb{N}_0}$. Nevertheless, we are able to prove the sublinear convergence rate for the suboptimality gap for subhomogeneous (Azé & Penot, 1995, p. 708) reference functions using a different technique based on generalized conjugacy. More precisely, we utilize the characterization of anisotropically smooth functions as (generalized) envelopes and interpret the algorithm as a nonlinear proximal point method, generalizing thus the duality between gradient descent and proximal point in the Euclidean case (Laude, 2021, Theorem 3.8). Then, we combine the subhomogeneity of $\phi$ with the proof technique of (Doikov & Nesterov, 2020, Theorem 1) for inexact tensor methods and obtain the claimed rate. This result is captured in the following theorem.

**Theorem 3.7.** *Let Assumption 3.1 hold, $f$ be convex and $\{x^k\}_{k\in\mathbb{N}_0}$ be the sequence of iterates generated by (2) with $\gamma = L^{-1}$, $\lambda = \bar{L}^{-1}$ and assume that $\mathrm{dom}\,\phi = \mathbb{R}^n$. Moreover, let $\phi$ be 2-subhomogeneous, i.e., such that $\phi(\theta x) \leq \theta^2\phi(x)$ for all $\theta \in [0, 1]$. Then, for all $K \geq 1$*

$$f(x^K) - f^\star \leq \frac{4\mathcal{D}_0}{K}, \qquad (12)$$

*where $\mathcal{D}_0 = \sup\{\bar{L}(L^{-1} \star \phi)(x - x^\star) : f(x) \leq f(x^0)\}$ for $x^\star \in \arg\min f$.*

In general the set $\mathcal{D}_0$ might be unbounded, except if $f$ has bounded level-sets, which is the case if $\arg\min f$ is bounded in light of (Bauschke et al., 2017b, Proposition 11.13). In theory, this dependence on the initial level-set can be eliminated by considering an averaging procedure akin to (Doikov & Nesterov, 2020, Algorithm 3), thus leading to a convergence rate in terms of some function of the initial distance to the solution. Nevertheless, such methods tend to underperform in practice compared to their more straightforward counterparts.

Examples of 2-subhomogeneous reference functions are those generated by $\cosh - 1$, as described in the following Lemma.

**Lemma 3.8.** *The function $h(x) := \cosh(x) - 1$ is 2-*

*subhomogeneous, i.e., the following inequality holds:*

$$h(\theta x) \leq \theta^2 h(x), \tag{13}$$

*for all $\theta \in [0,1]$ and $x \in \mathbb{R}$.*

## 4. Experiments

In this section we present some simple experiments that display the behavior of the proposed method on problems beyond traditional Lipschitzian assumptions. The code for reproducing the experiments is publicly available[1].

### 4.1. Norm to power

For the first part of our experiments we consider the toy example of minimizing $f(x) = \frac{1}{4}\|x\|^4$, with $x \in \mathbb{R}^{500}$, using different preconditioning schemes. We consider the reference functions $\cosh(\|x\|) - 1$, $\exp(\|x\|) - \|x\| - 1$ and $-\|x\| - \ln(1 - \|x\|)$ and remind that the algorithm generated by the latter function is a tighter version of the algorithm proposed in (Gorbunov et al., 2024). In this experiment, we keep $\bar{L}$ fixed, compute $L$ according to the rules established in Appendix D and apply algorithm (2) with $\gamma = L^{-1}$ and $\lambda = \bar{L}^{-1}$. The results are presented in Figure 2. For different values of $L$ and $\bar{L}$ there seems to be a trade-off between faster convergence to medium accuracy and slower convergence to very good accuracy for all three preconditioned methods.

### 4.2. Nonconvex phase retrieval

In this experiment we consider the nonconvex phase retrieval problem

$$\min_{x \in \mathbb{R}^n} f(x) = \frac{1}{2m} \sum_{i=1}^{m} (y_i - (a_i^\top x)^2)^2$$

with $y_i \in \mathbb{R}, a_i \in \mathbb{R}^n$. The data is generated as in (Chen et al., 2023): $n = 100$, $m = 3000$ and $a_i, z \sim \mathcal{N}(0, 0.5)$, $x_0 \sim \mathcal{N}(5, 0.5)$ generated element-wise with $z$ denoting the true underlying object. The measurements are generated as $y_i = (a_i^\top z)^2 + n_i$ with $n_i \sim \mathcal{N}(0, 4^2)$.

We compare the algorithm (2) with the isotropic and anisotropic reference functions generated by $\cosh - 1$, denoted respectively by $\phi_1(x) = \cosh(\|x\|) - 1$ and $\phi_2(x) = \sum_{i=1}^{n} \cosh(x_i) - 1$ against vanilla gradient descent, gradient clipping (Zhang et al., 2020b) and (Chen et al., 2023, Algorithm 1) with the tuning described in (Chen et al., 2023, Section 7). For the isotropic case of (2) we take $\gamma = 5/3$ and $\lambda = 1/100$, while for the anisotropic one $\gamma = 1/5$ and $\lambda = 1/14$. The results are presented in Figure 3. We use as $f^\star$ the minimum value of the cost function

---

[1] https://github.com/JanQ/
nonlinearly-preconditioned-gradient

among all algorithms. In this experiment the two versions of the algorithm proposed in this paper outperform the rest of the methods.

Moreover, we test the clipping and the isotropic algorithm over 100 random instances of the problem and plot the mean along with error bars representing one standard deviation on a logarithmic scale. It can be seen that the isotropic algorithm outperforms the clipping method across the tests for this particular tuning. Note that the tuning for both of the methods is quite robust.

### 4.3. Neural network training

In this experiment we consider training a simple four-layer fully connected network with layer dimensions [$28 \times 28, 128, 64, 32, 32, 10$] and ReLU activation functions on a subset of the MNIST dataset (Deng, 2012), using the cross-entropy loss. We consider a subset ($m = 600$) of the dataset in order to efficiently use full gradient updates.

We compare the methods generated by $\phi_1(x) = \cosh(\|x\|) - 1$, $\phi_2(x) = -\|x\| - \ln(1 - \|x\|)$ and the gradient clipping method (Zhang et al., 2020b), that can also be considered as an instance of (2) through Example 1.7, for various choices of the stepsizes and the clipping parameters. The results are presented in Figure 4. It can be seen that different combinations of $\gamma$ and $\lambda$ lead to different behaviors for the compared methods.

## 5. Conclusion and Future Work

In this paper we introduced and studied a new generalized smoothness inequality that is less restrictive than Lipschitz smoothness. We provided sufficient first- and second-order conditions through an unconventional technique that also leads to novel insights into the class of $(L_0, L_1)$-smooth functions. We moreover analyzed a nonlinearly preconditioned gradient scheme that is tailored to the proposed smoothness condition and studied its convergence properties both in the nonconvex and convex setting. This framework encapsulates a plethora of well-known methods, while it also generates new algorithms.

Our work paves the way for better understanding clipping and signed gradient methods from a majorization-minimization perspective. Possible interesting future work includes integrating momentum both in the convex and nonconvex regime and studying the stochastic setup. Another interesting research direction is extending our convergence results for the suboptimality gap from the smooth to the additive nonsmooth setting where the nonsmooth term is handled similarly to (Laude & Patrinos, 2025). We believe that this extension is not straightforward and requires additional effort compared to the standard Euclidean setup of gradient descent.

## Acknowledgements

Work supported by: the Research Foundation Flanders (FWO) research projects G081222N, G033822N, G0A0920N; Research Council KUL grant C14/24/103.

The authors thank Thomas Moellenhoff for helpful conversations.

## Impact statement

This paper presents work whose goal is to advance the field of Machine Learning. There are many potential societal consequences of our work, none which we feel must be specifically highlighted here.

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

## A. Missing proofs of Section 1

### A.1. Proof of Lemma 1.3

*Proof.* To begin with, since $h$ is proper, lsc and convex, $h = h^{**}$. In light of (Rockafellar & Wets, 1998, Theorem 11.8), we have that $\min h^* = -h^{**}(0) = -h(0) = 0$, implying that $h^* \geq 0$. Moreover, from the same theorem, $\arg\min h = \{h^{*\prime}(0)\}$ further implying that $h^{*\prime}(0) = 0$ and $h^*(0) = 0$. Since $h$ is even, we have from (Bauschke et al., 2017b, Example 13.8) that $h^* = (h \circ |\cdot|)^* = h^* \circ |\cdot|$, which means that $h^*$ is also even. Therefore, through (Bauschke et al., 2017b, Proposition 11.7) we get that $h^*$ is increasing on $\mathbb{R}_+$.

Now, note that the function $g = h - \frac{\mu}{2}|\cdot|^2$ is proper, lsc and convex where $\mu$ is the strong convexity parameter of $h$. Moreover, it is even as the difference of two even functions. Therefore, from (Bauschke et al., 2017b, Proposition 11.7) $g$ is an increasing function on $\mathbb{R}_+$ and thus $g \circ \|\cdot\|$ is a convex function on $\mathbb{R}^n$. This implies that $h(\|x\|) - \frac{\mu}{2}\|x\|^2$ is convex and thus that $\phi = h \circ \|\cdot\|$ is strongly convex with the same strong convexity parameter.

Again from (Bauschke et al., 2017b, Example 13.8), we have that $\phi^* = h^* \circ \|\cdot\|$ and to get the gradient of $\phi^*$ we can then utilize (Bauschke et al., 2017b, Corollary 16.72) to obtain $\nabla\phi^*(y) = h^{*\prime}(\|y\|)\overline{\mathrm{sgn}}(y)$.

Regarding the twice continuous differentiability of $\phi^*$, it follows from (Rockafellar, 1977, p. 42) that $h^* \in \mathcal{C}^2(\mathbb{R}^n)$ and the claimed result follows from (Strichartz, 2000, Exercise 10.2.20), since $h^*$ is even. $\qquad\square$

### A.2. Proof of Example 1.7

*Proof.* It is straightforward that $h(x) = \frac{1}{2}x^2 + \delta_{[-1,1]}(x)$, is a proper, lsc, strongly convex and even function with $h(0) = 0$. Then, from (Rockafellar & Wets, 1998, Theorem 11.23) we have that $h^*(y) = \inf_x \sigma_{[-1,1]}(x) + \frac{1}{2}(y - x)^2$, where $\sigma_{[-1,1]}$ is the support function of $[-1, 1]$ and in light of (Rockafellar & Wets, 1998, Exercise 11.27), $h^{*\prime}(y) = \Pi_{[-1,1]}(y)$, where $\Pi_{[-1,1]}(y) = \min(1, \max(-1, y))$ is the projection on the closed convex set $[-1, 1]$. Using Lemma 1.3 we thus obtain $\nabla\phi^*(y) = \min(1, \|y\|)\overline{\mathrm{sgn}}(y)$ and the algorithm becomes:

$$x^{k+1} = x^k - \gamma \min(1/\|\nabla f(x^k)\|, \lambda)\nabla f(x^k),$$

by pulling the norm inside the $\min$. $\qquad\square$

## B. Missing proofs of Section 2

### B.1. Proof of Proposition 2.2

*Proof.* Consider the following quantities, which are generalized conjugates as defined in (Rockafellar & Wets, 1998, Chapter 11L):

$$(-f)^\Phi(y) = \sup_{x \in \mathbb{R}^n} -\bar{L}(L^{-1} \star \phi)(x - y) + f(x) = -\inf_{x \in \mathbb{R}^n} \bar{L}(L^{-1} \star \phi)(x - y) - f(x),$$

$$(-f)^{\Phi\Phi}(x) = \sup_{y \in \mathbb{R}^n} -\bar{L}(L^{-1} \star \phi)(x - y) - (-f)^\Phi(y).$$

Let $\bar{x} \in \mathbb{R}^n$. Since $f$ is $(L, \bar{L})$-anisotropically smooth, we have that

$$\bar{x} \in \arg\min_{x \in \mathbb{R}^n} \bar{L}(L^{-1} \star \phi)(x - \bar{y}) - f(x),$$

where $\bar{y} = T_{L^{-1}, \bar{L}^{-1}}(\bar{x}) = \bar{x} - L^{-1}\nabla\phi^*(\bar{L}^{-1}\nabla f(\bar{x}))$ and thus

$$(-f)^\Phi(\bar{y}) = -\bar{L}(L^{-1} \star \phi)(\bar{x} - \bar{y}) + f(\bar{x}) \in \mathbb{R}, \qquad (14)$$

which implies that $(-f)^\Phi$ is proper. Since $\bar{x}$ was arbitrary, this holds for any $x \in \mathbb{R}^n$ and $y = T_{L^{-1}, \bar{L}^{-1}}(x)$. By the definition of $(-f)^{\Phi\Phi}$ for such pairs $x$ and $y$ we have that

$$(-f)^{\Phi\Phi}(x) + (-f)^\Phi(y) \geq -\bar{L}(L^{-1} \star \phi)(x - y).$$

Substituting (14) in the inequality above, we obtain $(-f)^{\Phi\Phi}(x) + f(x) \geq 0$ or $-f(x) \leq (-f)^{\Phi\Phi}(x)$. Moreover, by the definition of $(-f)^\Phi$, we have that for any $x, y \in \mathbb{R}^n$

$$(-f)^\Phi(y) - f(x) \geq -\bar{L}(L^{-1} \star \phi)(x - y).$$

Moving $(-f)^\Phi(y)$ to the other side and taking the supremum with respect to $y$ we obtain $-f(x) \geq (-f)^{\Phi\Phi}(x)$, which combined with the previous result means that $-f(x) = (-f)^{\Phi\Phi}(x)$. Therefore, we have

$$-f(x) = (-f)^{\Phi\Phi}(x) = \sup_{y \in \mathbb{R}^n} -\bar{L}(L^{-1} \star \phi)(x - y) - (-f)^\Phi(y)$$
$$= -\inf_{y \in \mathbb{R}^n} \bar{L}(L^{-1} \star \phi)(x - y) + (-f)^\Phi(y),$$

which is the claimed result for $\xi = (-f)^\Phi$.

We now show the convexity of $f^* - L^{-1}(\bar{L} \star \phi^*)$. In light of (Rockafellar & Wets, 1998, Theorem 11.23) and (Bauschke et al., 2017b, Proposition 13.23) we have

$$f^* = \left( \inf_{y \in \mathbb{R}^n} \bar{L}(L^{-1} \star \phi)(\cdot - y) + (-f)^\Phi(y) \right)^* = \xi^* + (\bar{L}(L^{-1} \star \phi))^* = \xi^* + L^{-1}(\bar{L} \star \phi^*)$$

and as such the result follows for $\psi = \xi^*$ which is lsc and convex, using moreover the fact that $\operatorname{dom} \phi^* = \mathbb{R}^n$ since $\phi$ is strongly convex. $\square$

## B.2. Proof of Proposition 2.3

*Proof.* Consider any points $x, \bar{x} \in \mathbb{R}^n$. If $L(x - \bar{x} + L^{-1}\nabla\phi^*(\nabla f(\bar{x}))) \notin \operatorname{dom} \phi$ then the bound holds trivially. Otherwise, from the Euclidean descent lemma for $f$ we have:

$$f(x) \leq f(\bar{x}) + \langle \nabla f(\bar{x}), x - \bar{x} \rangle + \frac{L_f}{2}\|x - \bar{x}\|^2.$$

Moreover, from the strong convexity of $\phi$ between points $L(x - \bar{x} + L^{-1}\nabla\phi^*(\nabla f(\bar{x})))$ and $\nabla\phi^*(\nabla f(\bar{x}))$:

$$\frac{1}{L} \star \phi(x - \bar{x} + L^{-1}\nabla\phi^*(\nabla f(\bar{x}))) = \frac{1}{L}\phi(L(x - \bar{x} + L^{-1}\nabla\phi^*(\nabla f(\bar{x}))))$$
$$\geq \frac{1}{L}\left[ \phi(\nabla\phi^*(\nabla f(\bar{x}))) + L\langle \nabla f(\bar{x}), x - \bar{x} \rangle + \frac{\mu L^2}{2}\|x - \bar{x}\|^2 \right]$$
$$= \frac{1}{L} \star \phi(L^{-1}\nabla\phi^*(\nabla f(\bar{x}))) + \langle \nabla f(\bar{x}), x - \bar{x} \rangle + \frac{\mu L}{2}\|x - \bar{x}\|^2,$$

where we have used the fact that $\operatorname{rge} \nabla\phi^* \subseteq \operatorname{dom} \phi$ along with $\nabla f(\bar{x}) \in \partial\phi(\nabla\phi^*(\nabla f(\bar{x})))$. Therefore, the claimed result follows. $\square$

## B.3. Proof of Lemma 2.5

*Proof.* Let $Q := \nabla^2 f(x)$ and note that $H := \nabla^2\phi^*(\bar{L}^{-1}\nabla f(x)) \succ 0$ from (Rockafellar, 1977, p. 42). Then

$$Q \prec \bar{L}LH^{-1} \iff H^{1/2}QH^{1/2} \prec \bar{L}LI \iff \lambda_{\max}(H^{1/2}QH^{1/2}) < \bar{L}L \iff \lambda_{\max}(HQ) < \bar{L}L,$$

where the first equivalence follows by (Horn & Johnson, 2012, Theorem 7.7.2c) with $S = H^{1/2}$. The last equivalence follows by noting that the (generally nonsymmetric) matrix $HQ$ is similar to the symmetric matrix $H^{1/2}QH^{1/2}$, by using (Horn & Johnson, 2012, Theorem 1.3.22) with $A = H^{1/2}Q$ and $B = H^{1/2}$, and noting that $H^{1/2}$ is nonsingular. The sufficient condition follows from Weyl's inequality (Horn & Johnson, 2012, Theorem 4.3.1).

$\square$

## B.4. Proof of Proposition 2.6

*Proof.* We will prove that $T_{L^{-1}, \bar{L}^{-1}} = \operatorname{id} - L^{-1}\nabla\phi^* \circ (\bar{L}^{-1}\nabla f)$ is a global homeomorphism from $\mathbb{R}^n \to \mathbb{R}^n$ using (Facchinei & Pang, 2003, Theorem 2.1.10).

Note that from Definition 2.4, $\lim_{\|x\| \to \infty} \|T_{L^{-1}, \bar{L}^{-1}}(x)\| = \infty$, i.e. $T_{L^{-1}, \bar{L}^{-1}}$ is norm-coercive, we only need to show that it is everywhere a local homeomorphism. The mapping $T_{L^{-1}, \bar{L}^{-1}}$ is locally Lipschitz, since $\nabla\phi^*$ is globally Lipschitz and $f \in \mathcal{C}^2(\mathbb{R}^n)$ and thus the generalized Jacobian is well-defined. Now, we have that $\partial_C T_{L^{-1}, \bar{L}^{-1}}(x) = \{I - L^{-1}V : V \in \partial_C(\nabla\phi^* \circ \bar{L}^{-1}\nabla f)(x)\}$ and in light of (Clarke, 1990, p. 75),

$$\partial_C(\nabla\phi^* \circ \bar{L}^{-1}\nabla f)(x)v \subseteq \operatorname{con}\{\partial_C(\nabla\phi^*)(\bar{L}^{-1}\nabla f(x))\bar{L}^{-1}\nabla^2 f(x)v\}$$

for any $v \in \mathbb{R}^n$. In order to show that $T_{L^{-1}, \bar{L}^{-1}}$ is everywhere a local homeomorphism we are going to use Clarke's inverse function theorem as presented in (Dontchev & Rockafellar, 2009, Theorem 4D.4). Consider thus any point $\bar{x} \in \mathbb{R}^n$ and the mapping $G_A(x) := T_{L^{-1}, \bar{L}^{-1}}(\bar{x}) + A(x - \bar{x})$, where $A \in \partial_C T_{L^{-1}, \bar{L}^{-1}}(\bar{x})$. Now, from the reasoning above we have that

$$A(x - \bar{x}) \in \mathrm{con}\{(I - L^{-1} \partial_C (\nabla \phi^*)(\bar{L}^{-1} \nabla f(\bar{x})) \bar{L}^{-1} \nabla^2 f(\bar{x}))(x - \bar{x})\}.$$

From Definition 2.4 we have that for all $H \in \partial_C (\nabla \phi^*)(\bar{L}^{-1} \nabla f(x))$, $\lambda_{\min}(I - L^{-1} \bar{L}^{-1} H \nabla^2 f(\bar{x})) > 0$, implying that $G_A$ is an invertible linear mapping for any $A \in \partial_C T_{L^{-1}, \bar{L}^{-1}}(\bar{x})$. Therefore, from (Dontchev & Rockafellar, 2009, Theorem 4D.4), there exists a Lipschitz continuous mapping $T_{L^{-1}, \bar{L}^{-1}}^{-1}$ such that $T_{L^{-1}, \bar{L}^{-1}}^{-1}(T_{L^{-1}, \bar{L}^{-1}}(x)) = x$ for some neighborhoods $U$ of $\bar{x}$ and $V$ of $T_{L^{-1}, \bar{L}^{-1}}(\bar{x})$. Since, moreover $T_{L^{-1}, \bar{L}^{-1}}$ is Lipschitz continuous, it is a local homeomorphism at $\bar{x}$ in the sense of (Facchinei & Pang, 2003, Definition 2.1.9). Since this holds for any $\bar{x} \in \mathbb{R}^n$, it is everywhere a local homeomorphism. This concludes our proof by using (Facchinei & Pang, 2003, Theorem 2.1.10).

$\square$

## B.5. Proof of Corollary 2.7

Consider $\phi(x) = -\|x\| - \ln(1 - \|x\|)$. In this case, $h(x) = -|x| - \ln(1 - |x|)$ and thus from Lemma 1.3, $\nabla \phi^*(y) = \frac{y}{1 + \|y\|}$. We thus have

$$\nabla^2 \phi^*(y) = \frac{1}{1 + \|y\|} I + \left( \frac{1}{(1 + \|y\|)^2} - \frac{1}{1 + \|y\|} \right) \frac{y y^\top}{\|y\|^2}$$
$$= \frac{1}{1 + \|y\|} \left[ I - \frac{\|y\|}{(1 + \|y\|)} \frac{y y^\top}{\|y\|^2} \right]$$

The term multiplying $\frac{y y^\top}{\|y\|^2}$ is negative and as such $\lambda_{\max}(\nabla^2 \phi^*(y)) \leq \frac{1}{1 + \|y\|}$. Moreover, $\nabla^2 \phi^*(y)$ is positive-definite, since $1 > \frac{\|y\|}{1 + \|y\|}$, which then implies $\|\nabla^2 \phi^*(y)\| \leq \frac{1}{1 + \|y\|}$. Therefore,

$$\|\nabla^2 \phi^*(\bar{L}^{-1} y)\| \leq \frac{\bar{L}}{\bar{L} + \|y\|} = \frac{L_0}{L_0 + L_1 \|y\|},$$

by choosing $\bar{L} = L_0 / L_1$. By $(L_0, L_1)$-smoothness we moreover have $\|\nabla^2 f(x)\| \leq L_0 + L_1 \|\nabla f(x)\|$ and thus

$$\|\nabla^2 \phi^*(\bar{L}^{-1} \nabla f(x)) \nabla^2 f(x)\| \leq \|\nabla^2 \phi^*(\bar{L}^{-1} \nabla f(x))\| \|\nabla^2 f(x)\| \leq L_0.$$

Choosing now $L = L_1$ we further have

$$\|\nabla^2 \phi^*(\bar{L}^{-1} \nabla f(x)) \nabla^2 f(x)\| \leq L \bar{L}. \tag{15}$$

Now take any points $x, \bar{x} \in \mathbb{R}^n$ and note that from the Cauchy–Schwarz inequality:

$$\langle \nabla \phi^*(\bar{L}^{-1} \nabla f(x)) - \nabla \phi^*(\bar{L}^{-1} \nabla f(\bar{x})), x - \bar{x} \rangle \leq \|\nabla \phi^*(\bar{L}^{-1} \nabla f(x)) - \nabla \phi^*(\bar{L}^{-1} \nabla f(\bar{x}))\| \|x - \bar{x}\|. \tag{16}$$

By the fundamental theorem of calculus for the mapping $\nabla \phi^* \circ (\bar{L}^{-1} \nabla f)$,

$$\nabla \phi^*(\bar{L}^{-1} \nabla f(x)) - \nabla \phi^*(\bar{L}^{-1} \nabla f(\bar{x})) = \int_0^1 \bar{L}^{-1} \nabla^2 \phi^*(\bar{L}^{-1} \nabla f(\bar{x} + t(x - \bar{x}))) \nabla^2 f(\bar{x} + t(x - \bar{x}))(x - \bar{x}) dt$$

and as such

$$\|\nabla \phi^*(\bar{L}^{-1} \nabla f(x)) - \nabla \phi^*(\bar{L}^{-1} \nabla f(\bar{x}))\| = \bar{L}^{-1} \left\| \int_0^1 \nabla^2 \phi^*(\nabla f(\bar{x} + t(x - \bar{x}))) \nabla^2 f(\bar{x} + t(x - \bar{x}))(x - \bar{x}) dt \right\|$$
$$\leq \bar{L}^{-1} \int_0^1 \|\nabla^2 \phi^*(\nabla f(\bar{x} + t(x - \bar{x}))) \nabla^2 f(\bar{x} + t(x - \bar{x}))\| dt \|x - \bar{x}\|$$
$$\leq L \|x - \bar{x}\|,$$

where the second inequality follows by (15). Putting this result back into (16), multiplying with $-\gamma < 0$ and adding $\|x - \bar{x}\|^2$ to both sides we obtain:

$$\|x - \bar{x}\|^2 - \langle \gamma \nabla \phi^*(\bar{L}^{-1} \nabla f(x)) - \gamma \nabla \phi^*(\bar{L}^{-1} \nabla f(\bar{x})), x - \bar{x} \rangle \geq (1 - \gamma L)\|x - \bar{x}\|^2,$$

implying that

$$\langle T_{\delta L^{-1}, \bar{L}^{-1}}(x) - T_{\delta L^{-1}, \bar{L}^{-1}}(\bar{x}), x - \bar{x} \rangle \geq (1 - \delta)\|x - \bar{x}\|^2,$$

since $\gamma = \delta L^{-1}$.

## B.6. Proof of Proposition 2.9

*Proof.* Note that (4) is equivalent to $\bar{x} \in \arg\min_{x \in \mathbb{R}^n} g(x) := \bar{L}(L^{-1} \star \phi)(x - \bar{y}) - f(x)$ for all $\bar{x} \in \mathbb{R}^n$, where $\bar{y} = \bar{x} - L^{-1} \nabla \phi^*(\bar{L}^{-1} \nabla f(\bar{x}))$. In light of the modern version of Fermat's theorem (Rockafellar & Wets, 1998, Theorem 10.1), for $\tilde{x}$ to be a local minimizer of $g$, the following inclusion should hold: $0 \in \widehat{\partial} g(\tilde{x})$. Through (Rockafellar & Wets, 1998, Exercise 10.10) this implies that

$$\nabla f(\tilde{x}) \in \partial (\bar{L}(L^{-1} \star \phi))(\tilde{x} - \bar{y}) = \bar{L} \partial \phi (L(\tilde{x} - \bar{y}))$$

or that $L(\tilde{x} - \bar{y}) = \nabla \phi^*(\bar{L}^{-1} \nabla f(\tilde{x}))$ or $\tilde{x} - L^{-1} \nabla \phi^*(\bar{L}^{-1} \nabla f(\tilde{x})) = \bar{y}$. This means that $T_{L^{-1}, \bar{L}^{-1}}(\tilde{x}) = T_{L^{-1}, \bar{L}^{-1}}(\bar{x})$ and since $T_{L^{-1}, \bar{L}^{-1}}$ is injective, the only possible minimizer of $g$ is $\bar{x}$.

Therefore, if we show that $g$ has a minimizer we are done.

**Case 1:** $\operatorname{dom} \phi$ is bounded. In light of (Rockafellar & Wets, 1998, p. 91), $g$ is a coercive function and as it is also proper and lsc, it attains its minimum.

**Case 2:** $\operatorname{dom} \phi = \mathbb{R}^n$. By the assumption of the proposition we have that

$$g(x) \geq \bar{L}(L^{-1} \star \phi)(x - \bar{y}) - \bar{L}(r^{-1} \star \phi)(x) + \beta. \tag{17}$$

Let $\mu$ be the strong convexity parameter of $\phi$, then we have the following for all $\alpha \in (0, 1)$:

$$
\begin{aligned}
\bar{L}(r^{-1} \star \phi)(x) &= \bar{L}r^{-1}\phi(rx) \\
&= \bar{L}r^{-1}\phi(r(x - \bar{y}) + r\bar{y}) \\
&= \bar{L}r^{-1}\phi\left(r\alpha \frac{x - \bar{y}}{\alpha} + r(1 - \alpha)\frac{\bar{y}}{1 - \alpha}\right) \\
&\leq \bar{L}r^{-1}\alpha\phi\left(\frac{r}{\alpha}(x - \bar{y})\right) + \bar{L}r^{-1}(1 - \alpha)\phi\left(\frac{r}{1 - \alpha}\bar{y}\right) - \frac{\mu}{2}\bar{L}r^{-1}\alpha(1 - \alpha)\left\|\frac{r}{\alpha}(x - \bar{y}) - \frac{r}{1 - \alpha}\bar{y}\right\|^2,
\end{aligned}
$$

where the inequality follows by the strong convexity inequality for $\phi$ between points $\frac{r}{\alpha}(x - \bar{y})$ and $\frac{r}{1-\alpha}\bar{y}$. Choosing now $\alpha = rL^{-1} < 1$ we obtain:

$$\bar{L}(r^{-1} \star \phi)(x) \leq \bar{L}(L^{-1} \star \phi)(x - \bar{y}) + \bar{L}(r^{-1}(1 - rL^{-1}) \star \phi)(\bar{y}) - \frac{\mu}{2} \frac{\bar{L}}{L(1 - rL^{-1})} \|(L - r)x - L\bar{y}\|^2.$$

Substituting this inequality in (17) we get

$$g(x) \geq -\bar{L}(r^{-1}(1 - rL^{-1}) \star \phi)(\bar{y}) + \frac{\mu}{2} \frac{\bar{L}}{L(1 - rL^{-1})} \|(L - r)x - L\bar{y}\|^2 + \beta =: \psi(x).$$

Note now that $\psi$ is a proper, lsc and strongly convex function and as such it has bounded level-sets. Due to the inequality, the level-sets of $g$ are contained in those of $\psi$ and thus $g$ also has bounded level-sets. Since moreover $g$ is lsc, it attains its minimum.

We thus have showed that in both of the above cases, $g$ has a minimizer and the proof is complete. $\square$

## C. Missing proofs of Section 3

### C.1. Proof of Theorem 3.2

*Proof.* From inequality (4) between points $x^{k+1}$ and $x^k$ we have:

$$f(x^{k+1}) \leq f(x^k) + \bar{L}L^{-1}[\phi((1-\alpha)\nabla\phi^*(\bar{L}^{-1}\nabla f(x^k))) - \phi(\nabla\phi^*(\bar{L}^{-1}\nabla f(x^k)))].$$

Using the fact that $\phi$ is even, we have $\phi((1-\alpha)\nabla\phi^*(\bar{L}^{-1}\nabla f(x^k))) = \phi(|1-\alpha|\nabla\phi^*(\bar{L}^{-1}\nabla f(x^k)))$ and since $\phi$ is convex, we have

$$\phi(\theta x) = \phi((1-\theta)0 + \theta x) \leq (1-\theta)\phi(0) + \theta\phi(x) = \theta\phi(x),$$

for any $\theta \in [0, 1]$. Note now that $|1 - \alpha| < 1$ and the previous inequality becomes:

$$f(x^{k+1}) \leq f(x^k) - (1 - |1 - \alpha|)\bar{L}L^{-1}\phi(\nabla\phi^*(\bar{L}^{-1}\nabla f(x^k))). \tag{18}$$

Therefore, summing up the above inequality we obtain

$$\sum_{k=0}^{K} \phi(\nabla\phi^*(\bar{L}^{-1}\nabla f(x^k))) \leq \frac{L}{\bar{L}\beta}(f(x^0) - f(x^{K+1})) \leq \frac{L}{\bar{L}\beta}(f(x^0) - f^\star),$$

which leads to

$$(K + 1) \min_{0 \leq k \leq K} \phi(\nabla\phi^*(\bar{L}^{-1}\nabla f(x^k))) \leq \frac{L}{\bar{L}\beta}(f(x^0) - f^\star). \tag{19}$$

Dividing now by $K + 1$ we obtain the claimed rate. $\qquad\square$

### C.2. Proof of Corollary 3.3

*Proof.* In light of Theorem 3.2, the following holds:

$$\min_{0 \leq k \leq K} \phi(\nabla\phi^*(\bar{L}^{-1}\nabla f(x^k))) \leq \frac{L(f(x^0) - f^\star)}{\bar{L}\beta(K + 1)}. \tag{20}$$

Now, using the fact that $\cosh(\text{arcsinh}(x)) = \sqrt{1 + x^2}$ we have:

$$\phi(\nabla\phi^*(\bar{L}^{-1}\nabla f(x^k))) = \cosh\left(\left\|\frac{\text{arcsinh}(\|\bar{L}^{-1}\nabla f(x^k)\|)}{\|\nabla f(x^k)\|}\nabla f(x^k)\right\|\right) - 1$$
$$= \cosh(\text{arcsinh}(\bar{L}^{-1}\|\nabla f(x^k)\|)) - 1$$
$$= \sqrt{1 + \bar{L}^{-2}\|\nabla f(x^k)\|^2} - 1.$$

The function $\sqrt{1 + x^2} - 1$ is increasing for $x \geq 0$ and as such $k^\star \in \arg\min_{0 \leq k \leq K}\{\phi(\nabla\phi^*(\bar{L}^{-1}\nabla f(x^k)))\}$ is equivalent to $k^\star \in \arg\min_{0 \leq k \leq K}\|\nabla f(x^k)\|^2$. Therefore, by taking 1 to the other side in (20) and taking the square, we obtain:

$$1 + \bar{L}^{-2}\|\nabla f(x^{k^\star})\|^2 \leq \left(\frac{L(f(x^0) - f^\star)}{\bar{L}\beta(K + 1)}\right)^2 + \frac{2L(f(x^0) - f^\star)}{\bar{L}\beta(K + 1)} + 1,$$

or that

$$\|\nabla f(x^{k^\star})\|^2 \leq \left(\frac{L(f(x^0) - f^\star)}{\beta(K + 1)}\right)^2 + \frac{2\bar{L}L(f(x^0) - f^\star)}{\beta(K + 1)}.$$

Taking the square root and using the fact that $\sqrt{\alpha + \beta} \leq \sqrt{\alpha} + \sqrt{\beta}$ we obtain the claimed result. $\qquad\square$

### C.3. Proof of Proposition 3.5

*Proof.* To begin with, with similar arguments as in the proof of Lemma 1.3 we have that $\phi^*(0) = 0$ and $\nabla\phi^*(0) = 0$. In light of Proposition 2.2, $f^* - L^{-1}(\bar{L} \star \phi^*)$ is a convex function. By definition $\nabla f(x) \in \operatorname{dom} \partial f^* \subseteq \operatorname{dom} f^*$ for all $x \in \mathbb{R}^n$ and as such we can consider the convex subgradient inequality for $f^* - L^{-1}(\bar{L} \star \phi^*)$ between points $\nabla f(x)$ and $\nabla f(x^\star)$ and obtain:

$$f^*(\nabla f(x)) - L^{-1}\bar{L}\phi^*(\bar{L}^{-1}\nabla f(x)) \geq f^*(\nabla f(x^\star)) + \langle x^\star, \nabla f(x)\rangle, \tag{21}$$

where we have used the fact that $\nabla f(x^\star) = 0$, $\nabla\phi^*(0) = 0$, $\phi^*(0) = 0$ and $x \in \partial f^*(\nabla f(x))$, since $f$ is convex. Taking once again the convex gradient inequality between points $\nabla f(x^\star)$ and $\nabla f(x)$, we now have:

$$f^*(\nabla f(x^\star)) \geq f^*(\nabla f(x)) - L^{-1}\bar{L}\phi^*(\bar{L}^{-1}\nabla f(x)) + \langle x - L^{-1}\nabla\phi^*(\bar{L}^{-1}\nabla f(x)), -\nabla f(x)\rangle, \tag{22}$$

with the same reasoning as before. Summing now (21) and (22) and rearranging we obtain the claimed result. $\qquad\square$

### C.4. Proof of Theorem 3.6

*Proof.* We begin as in the classical analysis of gradient descent by using the Pythagorean theorem:

$$\|x^{k+1} - x^\star\|^2 = \|x^k - x^\star\|^2 - 2L^{-1}\langle\nabla\phi^*(\bar{L}^{-1}\nabla f(x^k)), x^k - x^\star\rangle + \|L^{-1}\nabla\phi^*(\bar{L}^{-1}\nabla f(x^k))\|^2. \tag{23}$$

We further have:

$$\begin{aligned}
-L^{-1}\langle\nabla\phi^*(\bar{L}^{-1}\nabla f(x^k)), x^k - x^\star\rangle &= -L^{-1}\frac{h^{*\prime}(\|\bar{L}^{-1}\nabla f(x^k)\|)}{\|\bar{L}^{-1}\nabla f(x^k)\|}\langle\bar{L}^{-1}\nabla f(x^k), x^k - x^\star\rangle \\
&\leq -L^{-2}\frac{h^{*\prime}(\|\bar{L}^{-1}\nabla f(x^k)\|)}{\|\bar{L}^{-1}\nabla f(x^k)\|}\langle\bar{L}^{-1}\nabla f(x^k), \nabla\phi^*(\bar{L}^{-1}\nabla f(x^k))\rangle \\
&= -\|L^{-1}\nabla\phi^*(\bar{L}^{-1}\nabla f(x^k))\|^2,
\end{aligned} \tag{24}$$

where in the inequality we used Proposition 3.5 and in the equalities the fact that $\nabla\phi^*(L^{-1}\nabla f(x^k)) = \frac{h^{*\prime}(\|L^{-1}\nabla f(x^k)\|)}{\|L^{-1}\nabla f(x^k)\|}\bar{L}^{-1}\nabla f(x^k)$ from Lemma 1.3. In the inequality, we also used the fact that $h^{*\prime}(t) \geq 0$ for $t \geq 0$. Indeed, by convexity, we have that:

$$h^*(0) \geq h^*(t) - h^{*\prime}(t)t \iff h^{*\prime}(t)t \geq h^*(t) - h^*(0),$$

implying that $h^{*\prime}(t) \geq 0$ for all $t \geq 0$. Plugging now (24) into (23), we obtain

$$\|x^{k+1} - x^\star\|^2 \leq \|x^k - x^\star\|^2 - L^{-1}\langle\nabla\phi^*(\bar{L}^{-1}\nabla f(x^k)), x^k - x^\star\rangle \leq \|x^k - x^*\|^2 - \|L^{-1}\nabla\phi^*(\bar{L}^{-1}\nabla f(x^k))\|^2, \tag{25}$$

which shows the Fejér monotonicity of $\{x^k\}_{k\in\mathbb{N}_0}$ w.r.t. $x^\star \in \arg\min f$. Since $(\bar{L}h)^* = \bar{L} \star h^*$ and $h$ is an even function, we have that $(\bar{L}\phi)^* = \bar{L} \star (h^* \circ \|\cdot\|)$. In light of Proposition 2.2, we have that $f^* - L^{-1}(\bar{L}\phi)^*$ is a convex function. Now, for any $x, \bar{x} \in \mathbb{R}^n$, from the convex subgradient inequality for this function, between points $\nabla f(x) \in \operatorname{dom} \partial f^*$ and $\nabla f(\bar{x}) \in \operatorname{dom} \partial f^*$ we have:

$$(f^* - L^{-1}(\bar{L}\phi)^*)(\nabla f(x)) \geq (f^* - L^{-1}(\bar{L}\phi)^*)(\nabla f(\bar{x})) + \langle\bar{x} - \nabla(L^{-1}(\bar{L}\phi)^*)(\nabla f(\bar{x})), \nabla f(x) - \nabla f(\bar{x})\rangle,$$

where we have moreover used the fact that $\bar{x} \in \partial f^*(\nabla f(\bar{x}))$. Therefore,

$$\begin{aligned}
D_{(\bar{L}\phi)^*}(\nabla f(x), \nabla f(\bar{x})) &= (\bar{L}\phi)^*(\nabla f(x)) - (\bar{L}\phi)^*(\nabla f(\bar{x})) - \langle\nabla(\bar{L}\phi)^*(\nabla f(\bar{x})), \nabla f(x) - \nabla f(\bar{x})\rangle \\
&\leq L[f^*(\nabla f(x)) - f^*(\nabla f(\bar{x})) - \langle\bar{x}, \nabla f(x) - \nabla f(\bar{x})\rangle] \\
&= LD_f(\bar{x}, x)
\end{aligned}$$

where $D_g$ denotes the Bregman divergence associated with $g$ and the equality follows by the definition of the convex conjugate.

Thus, (Maddison et al., 2021, Assumption 3.1) holds for $k = (\bar{L}\phi)^*$. Therefore, substituting from (Maddison et al., 2021, Equation (41)), we obtain $(\bar{L}\phi)^*(\nabla f(x^{k+1})) \leq (\bar{L}\phi)^*(\nabla f(x^k))$ for all $k \in \mathbb{N}_0$ and since $h^*$ is an increasing function on $\mathbb{R}_+$ from Lemma 1.3,

$$\bar{L}h^*(\bar{L}^{-1}\|\nabla f(x^{k+1})\|) \leq \bar{L}h^*(\bar{L}^{-1}\|\nabla f(x^k)\|) \implies \|\nabla f(x^{k+1})\| \leq \|\nabla f(x^k)\|. \tag{26}$$

and thus we proved that the norm of the gradient of $f$ monotonically decreases along the iterates.

We now return to the Fejér-type inequality (25):

$$\|x^{k+1} - x^\star\|^2 \leq \|x^k - x^\star\|^2 - L^{-1}\frac{h^{*\prime}(\|\bar{L}^{-1}\nabla f(x^k)\|)}{\|\bar{L}^{-1}\nabla f(x^k)\|}\langle \bar{L}^{-1}\nabla f(x^k), x^k - x^\star\rangle.$$

Using the convex gradient inequality for $f$ we further have:

$$\|x^{k+1} - x^\star\|^2 \leq \|x^k - x^\star\|^2 - L^{-1}\bar{L}^{-1}\frac{h^{*\prime}(\|\bar{L}^{-1}\nabla f(x^k)\|)}{\|\bar{L}^{-1}\nabla f(x^k)\|}(f(x^k) - f^\star).$$

Summing up now the inequality above we obtain:

$$\sum_{k=0}^{K} L^{-1}\bar{L}^{-1}\frac{h^{*\prime}(\|\bar{L}^{-1}\nabla f(x^k)\|)}{\|\bar{L}^{-1}\nabla f(x^k)\|}(f(x^k) - f^\star) \leq \|x^0 - x^\star\|^2,$$

which after utilizing the fact that

$$\|\bar{L}^{-1}\nabla f(x^{k+1})\| \leq \|\bar{L}^{-1}\nabla f(x^k)\| \quad \forall k \in \mathbb{N}_0 \implies \frac{h^{*\prime}(\|\bar{L}^{-1}\nabla f(x^k)\|)}{\|\bar{L}^{-1}\nabla f(x^k)\|} \geq \frac{h^{*\prime}(\|\bar{L}^{-1}\nabla f(x^0)\|)}{\|\bar{L}^{-1}\nabla f(x^0)\|} \quad \forall k \in \mathbb{N}_0,$$

since $\frac{h^{*\prime}(x)}{x}$ is decreasing on $\mathbb{R}_+$, implies that

$$(K + 1)L^{-1}\frac{h^{*\prime}(\|\bar{L}^{-1}\nabla f(x^0)\|)}{\|\nabla f(x^0)\|}\min_{0\leq k\leq K}(f(x^k) - f^\star) \leq \|x^0 - x^\star\|^2.$$

This is the claimed result, since the function values decrease along the iterates of the algorithm from (18). $\qquad\square$

### C.5. Proof of Theorem 3.7

In light of Proposition 2.2, $f = \inf_{y\in\mathbb{R}^n} \bar{L}(L^{-1} \star \phi)(\cdot - y) + \xi(y)$ for some $\xi : \mathbb{R}^n \to \overline{\mathbb{R}}$. Since $f$ is moreover convex in this setting and $\operatorname{dom}\phi = \mathbb{R}^n$, we can take $\xi$ to be convex and lsc from (Laude & Patrinos, 2025, Proposition 4.1). In order to prove our result, we will resort to a nonlinear proximal point interpretation of (2) with a strongly convex prox-kernel. We therefore consider the following iteration:

$$x^{k+1} = \operatorname*{arg\,min}_{y\in\mathbb{R}^n} \bar{L}(L^{-1} \star \phi)(x^k - y) + \xi(y). \tag{27}$$

From (Laude & Patrinos, 2025, Proposition 3.9 (ii)), since $\operatorname{dom}\phi = \mathbb{R}^n$, we have that

$$\nabla f(x^k) = \nabla(\inf_{y\in\mathbb{R}^n} \bar{L}(L^{-1} \star \phi)(x^k - y) + \xi(y)) = \bar{L}\nabla\phi(L(x^k - x^{k+1})),$$

which directly implies that $x^{k+1} = x^k - L^{-1}\nabla\phi^*(\bar{L}^{-1}\nabla f(x^k))$ and the claimed equivalence between the two schemes is established. Using this result we can now prove a certain three-point-like property for the iterates generated by (2).

**Lemma C.1.** *Let $\{x^k\}_{k\in\mathbb{N}_0}$ be the sequence of iterates generated from* (27). *Then, the following inequality holds for all $x \in \mathbb{R}^n$:*

$$\xi(x^{k+1}) \leq \xi(x) - \bar{L}[(L^{-1} \star \phi)(x^k - x^{k+1}) - (L^{-1} \star \phi)(x^k - x)]. \tag{28}$$

*Proof.* By the optimality conditions for (27), we have that

$$0 \in -\bar{L}\nabla\phi(L(x^k - x^{k+1})) + \partial\xi(x^{k+1}).$$

Now, by the convex subgradient inequality for $\xi$, with $u^{k+1} \in \partial\xi(x^{k+1})$, we have that

$$\begin{aligned}
\xi(x) &\geq \xi(x^{k+1}) + \langle u^{k+1}, x - x^{k+1}\rangle \\
&= \xi(x^{k+1}) + \bar{L}\langle\nabla\phi(L(x^k - x^{k+1})), x - x^{k+1}\rangle \\
&= \xi(x^{k+1}) + \bar{L}L^{-1}\langle\nabla\phi(L(x^k - x^{k+1})), L(x - x^k) + L(x^k - x^{k+1})\rangle \\
&\geq \xi(x^{k+1}) + \bar{L}[(L^{-1} \star \phi)(x^k - x^{k+1}) - (L^{-1} \star \phi)(x^k - x)]
\end{aligned}$$

where the first equality follows by the inclusion above and the second by simple algebraic manipulations. The final inequality follows by using the convex gradient inequality for $\phi$ between points $L(x^k - x)$ and $L(x^k - x^{k+1})$. Therefore, the claimed result follows by rearranging. □

Now, we move on to the proof of our main theorem. It is inspired by the proof of (Doikov & Nesterov, 2020), where we have also utilized the fact that $\phi$ is 2-subhomogeneous.

*Proof.* As established above, we consider the sequence of iterates generated by (27). In light of Lemma C.1 and since $\phi \geq 0$ we have for any $x \in \mathbb{R}^n$:

$$\xi(x^{k+1}) \leq \xi(x) + \bar{L}(L^{-1} \star \phi)(x^k - x). \tag{29}$$

Now consider an arbitrary increasing sequence $\{A_k\}_{k \in \mathbb{N}_0}$ with $A_k > 0$ and $A_0 = 0$. We denote by $a_{k+1} := A_{k+1} - A_k$ and by taking $x := \frac{a_{k+1}x^\star + A_k x^k}{A_{k+1}}$ we have $x^k - x = \frac{a_{k+1}}{A_{k+1}}(x^k - x^\star)$. Plugging this in (29) and using the convexity of $\xi$ we obtain:

$$\xi(x^{k+1}) \leq \frac{a_{k+1}}{A_{k+1}}\xi(x^\star) + \frac{A_k}{A_{k+1}}\xi(x^k) + \bar{L}(L^{-1} \star \phi)(\frac{a_{k+1}}{A_{k+1}}(x^k - x^\star)).$$

Let $\theta_k := \frac{a_{k+1}}{A_{k+1}} \leq 1$. By the subhomogeneity of $\phi$ we have that

$$\xi(x^{k+1}) \leq \frac{a_{k+1}}{A_{k+1}}\xi(x^\star) + \frac{A_k}{A_{k+1}}\xi(x^k) + \theta_k^2 \bar{L}(L^{-1} \star \phi)(x^k - x^\star).$$

Multiplying both sides with $A_{k+1}$ we get since $a_{k+1} = A_{k+1} - A_k$

$$A_{k+1}\big(\xi(x^{k+1}) - \xi(x^\star)\big) \leq A_k\big(\xi(x^k) - \xi(x^\star)\big) + \frac{a_{k+1}^2}{A_{k+1}}\bar{L}(L^{-1} \star \phi)(x^k - x^\star).$$

Summing the inequality from $k = 0$ to $k = K - 1$ we obtain since $A_0 = 0$:

$$A_K\big(\xi(x^K) - \xi(x^\star)\big) \leq \sum_{k=0}^{K-1} \frac{a_{k+1}^2}{A_{k+1}}\bar{L}(L^{-1} \star \phi)(x^k - x^\star). \tag{30}$$

Choosing $x = x^k$ in (29) we have

$$\xi(x^{k+1}) \leq \xi(x^k).$$

and thus $\xi(x^K) - \xi(x^1) \leq 0$, which implies that

$$f(x^K) \leq \xi(x^K) \leq \xi(x^1) \leq \xi(x^1) + \bar{L}(L^{-1} \star \phi)(x^0 - x^1) = f(x^0).$$

The first inequality in the above display follows by the envelope representation of $f$, which implies that $f(x) \leq \xi(x)$ for all $x \in \mathbb{R}^n$. The equality also follows from the envelope representation, since

$$f(x^0) = \inf_{y \in \mathbb{R}^n} \bar{L}(L^{-1} \star \phi)(x^0 - y) + \xi(y) = \bar{L}(L^{-1} \star \phi)(x^0 - x^1) + \xi(x^1)$$

from (27). Thus we can further bound (30):

$$A_K\big(\xi(x^K) - \xi(x^\star)\big) \leq \mathcal{D}_0 \sum_{k=0}^{K-1} \frac{a_{k+1}^2}{A_{k+1}}.$$

We choose $A_k = k^2$ and by using the fact that $\sum_{k=1}^K \frac{a_k^2}{A_k} \leq 4K$ (Doikov & Nesterov, 2020, Equation (35)):

$$A_K\big(\xi(x^K) - \xi(x^\star)\big) \leq 4\mathcal{D}_0 K.$$

Dividing by $A_K$ we obtain:

$$\xi(x^K) - \xi(x^\star) \leq \frac{4\mathcal{D}_0}{K}.$$

Noting that $f(x^K) \leq \xi(x^K)$, from the envelope representation of $f$, and $\xi(x^\star) = f(x^\star)$ we obtain the desired result.

□

## C.6. Proof of Lemma 3.8

*Proof.* Fix $x \in \mathbb{R} \setminus \{0\}$ and consider the function $g(\theta) := \cosh(\theta x) - 1 - \theta^2(\cosh(x) - 1)$. If this function is at most nonpositive for $\theta \in [0, 1]$, then the claim is proven. Note that $g(0) = 0$ and $g(1) = 0$. Moreover, $g'(\theta) = x \sinh(\theta x) - 2\theta(\cosh(x) - 1)$ and thus $g'(0) = 0$. Now, $g''(\theta) = x^2 \cosh(\theta x) - 2 \cosh(x) + 2$ and thus $g''(0) = x^2 + 2 - 2 \cosh(x) < 0$, which further implies that 0 is a local maximum. Therefore, there exists a $\bar{\theta} \in (0, 1]$ such that $g(\theta) < g(0) = 0$ for all $\theta \in [0, \bar{\theta})$, which implies that if we prove that $g(\theta) \neq 0$ for all $\theta \in (0, 1)$ we are done.

Let us assume now that there exists a $\theta^* \in (0, 1)$ such that $g(\theta^*) = 0$. Then, by Rolle's theorem, there must exist two critical points for $g$ in $(0, 1)$, one in $(0, \theta^*)$ and one in $(\theta^*, 1)$. We have that

$$g'(\theta) = 0 \iff \sinh(\theta x) = 2\theta \frac{\cosh(x) - 1}{x}.$$

Setting $y = \theta x$ the equation above is the same as

$$\sinh(y) = 2 \frac{\cosh(x) - 1}{x^2} y = \alpha y,$$

which has exactly three solutions: $y_1 < 0$, $y_2 = 0$, $y_3 > 0$, since $2 \cosh(x) > 2 + x^2$ for $x \neq 0$. Without loss of generality we assume that $x > 0$ and thus we get that there exists only one $\theta > 0$ such that $g'(\theta) = 0$, which is a contradiction. $\qquad \square$

# D. Details on the second-order condition

In this section we provide further details on the second-order condition Definition 2.4. We complement the discussion in Section 2 by showing that the norm-coercivity condition on the forward operator in Definition 2.4 is in fact mild even for functions $\phi$ with full domain.

**Proposition D.1.** *Let $\phi = h \circ \|\cdot\|$ such that $h \in \mathcal{C}^1(\mathbb{R})$ satisfies Assumption 1.1. If there exists some $C > 0$ such that*

$$\|\nabla f(x)\| \leq \bar{L}|h'(L\|x\|)| \tag{31}$$

*for all $x$ such that $\|x\| \geq C$, then*

$$\lim_{\|x\| \to \infty} \|T_{\delta L^{-1}, \bar{L}^{-1}}(x)\| = \infty,$$

*for all $\delta < 1$.*

*Proof.* In the following we assume that $\|x\|$ is large enough such that the assumption of the proposition holds. We have that

$$\bar{L}^{-1}\|\nabla f(x)\| \leq |h'(L\|x\|)| \Rightarrow h^{*'}(\bar{L}^{-1}\|\nabla f(x)\|) \leq L\|x\|. \tag{32}$$

The implication follows since $h(0) \geq h(t) - h'(t)t$, meaning that $h'(t) \geq 0$ for $t \geq 0$ and thus $|h'(t)| = h'(t)$ on this interval implying $h^{*'}(|h'(t)|) = t$. Now, by the reverse triangle inequality:

$$\|T_{\delta L^{-1}, \bar{L}^{-1}}(x)\| \geq \|x\| - \delta L^{-1}|h^{*'}(\bar{L}^{-1}\|\nabla f(x)\|)|$$
$$\geq \|x\|(1 - \delta),$$

where the second inequality follows by (32). Therefore, since $\delta < 1$, $\lim_{\|x\| \to \infty} \|T_{\delta L^{-1}, \bar{L}^{-1}}(x)\| = \infty$ and the proof is complete. $\qquad \square$

The fact that this condition is quite mild can be seen by choosing $\phi(x) = \cosh(\|x\|) - 1$, where we allow $\|\nabla f(x)\|$ to grow exponentially with $\|x\|$. It is straightforward that this condition holds for example when the norm of the gradient is bounded by some polynomial of the norm of $x$ when $\|x\|$ is large enough.

We next show that when the matrix $H\nabla^2 f(x)$ is symmetric, the norm-coercivity property of the forward operator in Definition 2.4 is not required in order to obtain a result similar to Proposition 2.6.

**Proposition D.2.** *Let $f \in \mathcal{C}^2(\mathbb{R}^n)$ be such that for all $x \in \mathbb{R}^n$ and $H \in \partial_C(\nabla\phi^*)(\bar{L}^{-1}\nabla f(x))$,*

$$\lambda_{\max}(H\nabla^2 f(x)) \leq L\bar{L} \tag{33}$$

*and $H\nabla^2 f(x)$ is symmetric. Then, the following inequality holds:*

$$\langle T_{\gamma,\bar{L}^{-1}}(x) - T_{\gamma,\bar{L}^{-1}}(\bar{x}), x - \bar{x} \rangle \geq (1 - \gamma L)\|x - \bar{x}\|^2, \tag{34}$$

*for all $x, \bar{x} \in \mathbb{R}^n$. In particular, for $\gamma < L^{-1}$, the forward operator is strongly monotone with parameter $1 - \gamma L$ and thus injective.*

*Proof.* Note that the mapping $\nabla\phi^* \circ (\bar{L}^{-1}\nabla f)$ is locally Lipschitz, since $\nabla\phi^*$ is globally Lipschitz and $f \in \mathcal{C}^2(\mathbb{R}^n)$. Then, we can invoke the generalized mean value theorem in its summation form (Facchinei & Pang, 2003, Proposition 7.1.16): for two points $x, \bar{x} \in \mathbb{R}^n$, there exist $n$ points $z_i \in (x, \bar{x})$ and $n$ scalars $\alpha_i \geq 0$ summing to unity such that

$$\nabla\phi^*(\bar{L}^{-1}\nabla f(x)) - \nabla\phi^*(\bar{L}^{-1}\nabla f(\bar{x})) = \sum_{i=1}^{n} \alpha_i V_i(x - \bar{x}), \tag{35}$$

where $V_i \in \partial_C(\nabla\phi^* \circ \bar{L}^{-1}\nabla f)(z_i)$. Now, in light of (Clarke, 1990, p. 75),

$$\partial_C(\nabla\phi^* \circ \bar{L}^{-1}\nabla f)(x)v \subseteq \mathrm{con}\{\partial_C(\nabla\phi^*)(\bar{L}^{-1}\nabla f(x))\bar{L}^{-1}\nabla^2 f(x)v\}$$

for any $v \in \mathbb{R}^n$. Therefore, any $V_i(x - \bar{x})$ can be written as $\bar{L}^{-1}\sum_{j=1}^{d} \beta_j H_j \nabla^2 f(z_i)(x - \bar{x})$, for $d > 0$, with $H_j \in \partial_C(\nabla\phi^*)(\bar{L}^{-1}\nabla f(z_i))$ and $\beta_j \geq 0$ summing to unity.

Taking an inner product with $(x - \bar{x})$, we have that

$$\langle \nabla\phi^*(\bar{L}^{-1}\nabla f(x)) - \nabla\phi^*(\bar{L}^{-1}\nabla f(\bar{x})), x - \bar{x} \rangle = \bar{L}^{-1}\sum_{i=1}^{n} \alpha_i \sum_{j=1}^{d} \beta_j \langle x - \bar{x}, H_j \nabla^2 f(z_i)(x - \bar{x}) \rangle$$

$$\leq \bar{L}^{-1}\sum_{i=1}^{n} \alpha_i \bar{L}L\|x - \bar{x}\|^2$$

$$= L\|x - \bar{x}\|^2,$$

where we have used the fact that $\lambda_{\max}(H_j \nabla^2 f(z_i)) \leq L\bar{L}$.

By multiplying the inequality above with $-\gamma < 0$ and then adding $\|x - \bar{x}\|^2$ to both sides we obtain:

$$\|x - \bar{x}\|^2 - \langle \gamma\nabla\phi^*(\bar{L}^{-1}\nabla f(x)) - \gamma\nabla\phi^*(\bar{L}^{-1}\nabla f(\bar{x})), x - \bar{x} \rangle \geq (1 - \gamma L)\|x - \bar{x}\|^2,$$

implying that

$$\langle T_{\gamma,\bar{L}^{-1}}(x) - T_{\gamma,\bar{L}^{-1}}(\bar{x}), x - \bar{x} \rangle \geq (1 - \gamma L)\|x - \bar{x}\|^2,$$

which is the claimed result. $\qquad\square$

### D.1. Examples

We now move on to providing examples of functions satisfying Definition 2.4. We consider the reference functions $\phi_1(x) = \cosh(\|x\|) - 1$, $\phi_2(x) = \exp(\|x\|) - \|x\| - 1$ and $\phi_3(x) = -\|x\| - \ln(1 - \|x\|)$, which are generated by the (1-dimensional) kernel functions $h_1(x) = \cosh(x) - 1$, $h_2(x) = \exp(|x|) - |x| - 1$ and $h_3(x) = -|x| - \ln(1 - |x|)$.

We first recall the preconditioner $\nabla\phi^*$ and its Jacobian $\nabla^2\phi^*$ for general isotropic reference functions.

$$\nabla\phi_i^*(y) = h_i^{*\prime}(\|y\|)\overline{\mathrm{sgn}}(y) \qquad \forall y \in \mathbb{R}^n$$

and

$$\nabla^2\phi_i^*(y) = h_i^{*\prime\prime}(\|y\|)\frac{yy^\top}{\|y\|^2} + \frac{h_i^{*\prime}(\|y\|)}{\|y\|}\left(I - \frac{yy^\top}{\|y\|^2}\right) \qquad \forall y \in \mathbb{R}^n \setminus \{0\}$$

and $\nabla^2\phi_i^*(y) = h_i^{*\prime\prime}(\|y\|)I$ otherwise. For ease of presentation we denote $a_i(y) = h_i^{*\prime\prime}(\|y\|)$ and $b_i(y) = \frac{h_i^{*\prime}(\|y\|)}{\|y\|}$.

Table 2. Anisotropic smoothness constants for the examples of Appendix D.1

|  | $\phi_1$ | $\phi_2$ | $\phi_3$ |
|---|---|---|---|
| $L_{\mathrm{norm}}$ | $\dfrac{2^{1/3}\sqrt{3}}{\bar{L}^{1/3}}$ | $\dfrac{2^{2/3}}{\bar{L}^{1/3}}$ | $\dfrac{2^{4/3}}{3\bar{L}^{1/3}}$ |
| $L_{\mathrm{logistic}}$ | $\dfrac{\|\alpha\|^2}{\sqrt{16\bar{L}^2+\|\alpha\|^2}}$ | $\dfrac{\|\alpha\|^2}{4\bar{L}+\|\alpha\|}$ | $\dfrac{\bar{L}\|\alpha\|^2+\|\alpha\|^3}{4(\bar{L}+\|\alpha\|)^2}$ |

*Example* D.3 (Norm to power). Let $f(x)=\frac{1}{4}\|x\|^4$. Then, $f$ is $(L,\bar{L})$-anisotropically smooth relative to $\phi_i$ for any $\bar{L}>0$ and $L>L_{\mathrm{norm}}$ from the first row of Table 2.

*Proof.* We first consider the more general $f(x)=\frac{1}{p}\|x\|^p$ with $p\geq 4$. The gradient and Hessian of $f$ are given respectively by

$$\nabla f(x)=\|x\|^{p-2}x,\qquad \nabla^2 f(x)=\|x\|^{p-2}I+(p-2)\|x\|^{p-4}xx^\top. \tag{36}$$

The second-order condition then involves the following quantity:

$$\nabla^2\phi_i^*(\bar{L}^{-1}\nabla f(x))=a_i(\bar{L}^{-1}\|\nabla f(x)\|)\frac{\nabla f(x)\nabla f(x)^\top}{\|\nabla f(x)\|^2}+b_i(\bar{L}^{-1}\|\nabla f(x)\|)\left(I-\frac{\nabla f(x)\nabla f(x)^\top}{\|\nabla f(x)\|^2}\right)$$

$$=a_i(\bar{L}^{-1}\|x\|^{p-1})\frac{xx^\top}{\|x\|^2}+b_i(\bar{L}^{-1}\|x\|^{p-1})\left(I-\frac{xx^\top}{\|x\|^2}\right).$$

We thus have:

$$\nabla^2\phi_i^*(\bar{L}^{-1}\nabla f(x))\nabla^2 f(x)=(p-1)a_i(\bar{L}^{-1}\|x\|^{p-1})\|x\|^{p-2}\frac{xx^\top}{\|x\|^2}+b_i(\bar{L}^{-1}\|x\|^{p-1})\|x\|^{p-2}\left(I-\frac{xx^\top}{\|x\|^2}\right).$$

The largest eigenvalue of this (symmetric) matrix is

$$\lambda_{\max}(\nabla^2\phi_i^*(\bar{L}^{-1}\nabla f(x))\nabla^2 f(x))=\max\left\{(p-1)a_i(\bar{L}^{-1}\|x\|^{p-1}),b_i(\bar{L}^{-1}\|x\|^{p-1})\right\}\|x\|^{p-2}. \tag{37}$$

and for all the reference functions we consider in this subsection, $(p-1)a_i(\bar{L}^{-1}\|x\|^{p-1})\geq b_i(\bar{L}^{-1}\|x\|^{p-1})$. Therefore the inequality (5) dictates $(p-1)a_i(\bar{L}^{-1}\|x\|^{p-1})\|x\|^{p-2}<L\bar{L}$ for all $x\in\mathbb{R}^n$.

Now we specialize to each of the reference functions we consider as well as take $p=4$.

For $\phi_1$, $\phi_2$ and $\phi_3$, we obtain the conditions

$$\phi_1:\ \frac{3\|x\|^2}{\sqrt{\bar{L}^2+\|x\|^6}}<L,\qquad \phi_2:\ \frac{3\|x\|^2}{\bar{L}+\|x\|^3}<L,\qquad \phi_3:\ \frac{3\bar{L}\|x\|^2}{(\bar{L}+\|x\|^3)^2}<L$$

for which the left-hand sides are maximized at $\|x\|=(2\bar{L}^2)^{1/6}$, $\|x\|=(2\bar{L})^{1/3}$ and $\|x\|=(\frac{\bar{L}}{2})^{1/3}$ respectively. Plugging these values in yields the resulting lower bounds $L_{\mathrm{norm}}$ in Table 2.

Since in every case $H\nabla^2 f(x)$ is a symmetric matrix, it follows from Proposition D.2 that the operator $T_{\delta L^{-1},\bar{L}^{-1}}$ is injective for any $\delta<1$. This implies the anisotropic smoothness of $f$ relative to $\phi_3$ in light of Proposition 2.9 since $\mathrm{dom}\,\phi_3$ is bounded. For $\phi_1$ and $\phi_2$ the result follows from Proposition 2.9 by the reasoning in Remark 2.10. $\square$

We now move on to the logistic loss function.

*Example* D.4. Let $f(x)=\log(1+\exp(-\alpha^\top x))$. Then, $f$ is $(L,\bar{L})$-anisotropically smooth relative to $\phi_i$ for any $\bar{L}>0$ and $L>L_{\mathrm{logistic}}$ defined in the second row of Table 2.

*Proof.* The gradient and the hessian of $f$ are given respectively by

$$\nabla f(x)=-\frac{\alpha}{1+\exp(\alpha^\top x)},\qquad \nabla^2 f(x)=\frac{\alpha\alpha^\top}{\exp(-\alpha^\top x)(1+\exp(\alpha^\top x))^2}. \tag{38}$$

In this case the second-order condition becomes

$$a_i(\bar{L}^{-1}\nabla f(x))\frac{\|\alpha\|^2}{\exp(-\alpha^\top x)(1+\exp(\alpha^\top x))^2} < L\bar{L}. \tag{39}$$

The results from Table 2 for $\phi_1$, $\phi_2$ and $\phi_3$ then follow respectively from

$$\frac{\|\alpha\|^2}{(1+\exp(-\alpha^\top x))\sqrt{\bar{L}^2(1+\exp(\alpha^\top x))^2+\|\alpha\|^2}} \leq \frac{\|\alpha\|^2}{\sqrt{16\bar{L}^2+\|\alpha\|^2}} < L,$$

$$\frac{\|\alpha\|^2}{(\bar{L}(1+\exp(\alpha^\top x))+\|\alpha\|)(1+\exp(-\alpha^\top x))} \leq \frac{\|\alpha\|^2}{4\bar{L}+\|\alpha\|} < L,$$

$$\frac{\bar{L}\|\alpha\|^2}{(\bar{L}(1+\exp(\alpha^\top x))+\|\alpha\|)^2\exp(-\alpha^\top x)} \leq \frac{\bar{L}\|\alpha\|^2+\|\alpha\|^3}{4(\bar{L}+\|\alpha\|)^2} < L.$$

Note that in this case as well, $H\nabla^2 f(x)$ is a symmetric matrix and it follows from Proposition D.2 that the operator $T_{\delta \bar{L}^{-1},\bar{L}^{-1}}$ is injective for any $\delta < 1$. The growth condition in Proposition 2.9 is satisfied automatically, since $f$ is bounded. $\qquad\square$

## D.2. Gradient clipping

We now consider the case that $\phi = \frac{1}{2}\|\cdot\|^2 + \delta_{\overline{\mathbb{B}}(0,1)}(\cdot)$. By (Themelis et al., 2019, p. 404), the generalized Jacobian of the preconditioner is given by

$$\partial_C(\nabla\phi^*)(y) = \begin{cases} \{I\}, & \text{if } \|y\| < 1, \\ \text{con}\{I, \Pi_{y^\perp}\}, & \text{if } \|y\| = 1, \\ \{\|y\|^{-1}\Pi_{y^\perp}\}, & \text{if } \|y\| > 1. \end{cases}$$

where $\Pi_{y^\perp}$ denotes the projection matrix onto the orthogonal complement of the subspace spanned by $y$. The second-order condition (5) then becomes

$$\begin{cases} \lambda_{\max}(\nabla^2 f(x)) < L\bar{L}, & \text{if } \|\nabla f(x)\| < \bar{L}, \\ \lambda_{\max}\big(\alpha\nabla^2 f(x) + (1-\alpha)\bar{L}\Pi_{\nabla f(x)^\perp}\nabla^2 f(x)\big) < L\bar{L}, & \text{if } \|\nabla f(x)\| = \bar{L}, \\ \lambda_{\max}(\Pi_{\nabla f(x)^\perp}\nabla^2 f(x)) < L\|\nabla f(x)\|, & \text{if } \|\nabla f(x)\| > \bar{L}, \end{cases}$$

for all $\alpha \in [0,1]$. In particular, for $\alpha = 1$, the second condition becomes $\nabla^2 f(x) \prec L\bar{L}I$, and using the fact that $\lambda_{\max}(\Pi_{\nabla f(x)^\perp}) = 1$, we also have $\lambda_{\max}((\alpha I + (1-\alpha)\Pi_{\nabla f(x)^\perp})\nabla^2 f(x)) \leq \lambda_{\max}(\nabla^2 f(x))$ such that this clause can be merged with the first case. To rewrite the last case in terms of symmetric matrices, we note that $\lambda_{\max}(\Pi_{\nabla f(x)^\perp}\nabla^2 f(x)) = \lambda_{\max}(\Pi^2_{\nabla f(x)^\perp}\nabla^2 f(x)) = \lambda_{\max}(\Pi_{\nabla f(x)^\perp}\nabla^2 f(x)\Pi_{\nabla f(x)^\perp})$ where the last equality follows by (Horn & Johnson, 2012, Theorem 1.3.22). The second-order condition is therefore

$$\begin{cases} \nabla^2 f(x) \prec L\bar{L}I, & \text{if } \|\nabla f(x)\| \leq \bar{L}, \\ \Pi_{\nabla f(x)^\top}\nabla^2 f(x)\Pi_{\nabla f(x)^\top} \prec L\|\nabla f(x)\|I, & \text{if } \|\nabla f(x)\| > \bar{L}. \end{cases}$$

Note that the first case is akin to standard $L$-smoothness while the second case is reminiscent of $(L_0, L_1)$-smoothness with $L_0 = 0$ but restricted to some subspace. In particular, if $\nabla^2 f(x) = C(x)\nabla f(x)\nabla f(x)^\top$ for some $C : \mathbb{R}^n \to \mathbb{R}$, then the second case is always satisfied, and we only require that $\nabla^2 f(x) \prec L\bar{L}I$ for $\|\nabla f(x)\| \leq \bar{L}$. This is the case for both of the examples considered in Appendix D.1.

