# OpenReview forum: "Nonlinearly Preconditioned Gradient Methods under Generalized Smoothness"
_ICML.cc/2025/Conference — ICML 2025 oral_

### Official Review · Reviewer_ZfYL · 2025-02-22

**Overall Recommendation:** 3

**Summary:**

This work studies nonlinearly preconditioned gradient methods. After comparison with existing methods and discussing the existing connections, an extension of anisotropic smoothness was proposed. For specific choices of the reference function (1- isotropic reference function and 2- separable reference function) the procedure of calculating a second order condition for $(L,\bar{L})$-anisotropic smoothness was explained. Later, the convergence of the preconditioned gradient method (2) was studied under the settings: a) nonconvexity + smoothness and b) convexity. Simple experiments were conducted to evaluate the performance of the proposed method.

**Claims And Evidence:**

Most of the claims are supported. One confusing part is in line 101: "... we describe a common nonlinear gradient preconditioning scheme for the main iterates of popular algorithms including gradient clipping, Adam, Adagrad and ...". However, as described in Section 1.6, a special case of these methods were recovered. These special cases have no gradient accumulation or momentum which are important building blocks for the successful performance of the mentioned algorithms.

**Essential References Not Discussed:**

None

**Experimental Designs Or Analyses:**

Yes, the current experiments are sound. One  issue, is as follows:
The paper claims that their framework brings out novel insights for the class of ($L_0$-$L_1$)-smooth functions and it allows them to go beyond already established methods. While I appreciate such direction, the paper leaves me with no numerical insights toward understanding how this claim looks like in practice.

**Methods And Evaluation Criteria:**

Yes

**Other Comments Or Suggestions:**

I believe the paper would benefit from a comprehensive revision specially in terms of presentation and delivering the message. In many instances, vague statements leave the reader helpless. Such statements damage the independentness of the paper. Examples are:

line 255: "... are Legndre through (Laude & Patrinos, 2022, Proposition 4.1)"

line 258: "... when $\phi$ is a regular optimal transport cost in light of (Villani,2008 Definition 12.27)"

line 269: " Note that in this case (Gorbunov et al., 2024, Algorithm 1) can be viewed as (2) with a convservative ..."

line 272: "In a similar manner, the stepsize introduced in [Vankov et al., 2024, Equation (3.5)] can be viewed as a less tight version of Corollary 2.7."

line 379: "wile also answering the question posed in (Maddison et al., 2021, p. 17)."

Figure 3 (right): I suggest a probability bound instead of 100 instances since it would be a more meaningful visualization.

**Other Strengths And Weaknesses:**

Strengths: I like the idea of moving toward unified models to understand and gain insights about the existing concepts. This work endeavours to move toward this direction by relating to generalized and more realistic smoothness assumptions. They also explored the performance of algorithm (2) in terms of convergence rate under smooth nonconvex and convex assumptions. Unfortunately, this extension was not formulated numerically and experimentally, leaving the readers doubtful about the applicability of their $(L,\bar{L})$-smoothness extension.

**Questions For Authors:**

First, I'd like to thank the authors for their research.

1- I like the idealogy/motivation presented in lines 80-95. Why authors did not use this idealogy in practice and see how reference functions with appealing properties (like strong convexity) perform in training problems?

2- lines 417-419: Is this why Fig3(right) acieves slighly better results?

3- Why does the paper claims encompassing Adagrad, Adam, and Gradient Clipping methods, when they only recover a special case of  these methods? (On the side: the momentum plays a crucial role in the success of algorithms like Adam and should not be neglected.)

**Relation To Broader Scientific Literature:**

One of the contributions relates to an extension of the anisotropic smoothness. The new definition allows for two different constants to play a complementary role. Accordingly, a second order sufficient condition was defined. Additionally, the analysis for nonconvex functions nonsmooth functions was extended to nonconvex smooth functions. The new result allowed for larger step sizes compared to the previous work. The authors claim that they strengthen the results from some previous works in Theorem 3.6, but it was not mentioned that in what sense did they strengthen the prior results?

**Theoretical Claims:**

No

---

> ### Author Rebuttal · Authors · 2025-03-31
>
> We thank the reviewer for their feedback.
>
> Regarding the experimental design,  our paper is focused on theoretical analysis and thus we only provide preliminary experimental results on problems that are associated with generalized smoothness in the related literature (Chen et al., 2023; Gorbunov et al., 2024; Vankov et al., 2024).
>
> Moreover, in Theorem 3.6 we strengthen the results of (Laude & Patrinos, 2022) and  (Maddison et al., 2021) in the following ways: we show that for isotropic reference functions the sequence of iterates generated by the method is Fejér monotone and we obtain a sublinear rate for the suboptimality gap $f(x^k) - f^\star$. To the best of our knowledge such results have not been shown before.
>
> Regarding the presentation of our paper, we agree that we can make some statements more self-contained and we will change this in a revised version of the paper. We would also like to point out that the quoted text contains typos that are not present in the paper.
>
> The meaning of **Questions 1 & 2** is not entirely clear to us. In the following we answer the questions as we understood them. If the answers are not satisfactory, please provide some more clarification.
>
> Answer to **Question 1**:
>
> As already mentioned, in this paper we focus on theoretical analysis. Nevertheless, we believe that applying the nonlinearly preconditioned gradient method to training problems is interesting future work.
>
> Answer to **Question 2**:
>
> We conjecture that the algorithm generated by the reference function $\cosh-1$ outperforms the rest of the methods on our experiments because the preconditioner is better tailored to such high order polynomial functions.
>
> Answer to **Question 3**:
>
> We agree with the reviewer that our wording might cause confusion and we will change our text. What we show in Subsection 1.6 is that if one was to remove the gradient accumulation and momentum mechanisms from these methods, they would obtain the shown basic forward iterations. We are well aware that momentum plays a crucial role in the success of methods such as Adam and Adagrad. However, in this paper we mostly focused on laying the foundation for a new framework for analyzing various gradient descent type methods. Extensions such as adding momentum to recover schemes like Adam and Adagrad are possible and an exciting direction for future research. Moreover, through this connection, practical new algorithms can be derived by combining the momentum mechanisms with other reference functions $\phi$.
>
> ---
>
> References:
>
> Chen, Z., Zhou, Y., Liang, Y., and Lu, Z. Generalized-smooth nonconvex optimization is as efficient as smooth nonconvex optimization. In International Conference on Machine Learning, pp. 5396–5427. PMLR, 2023.
>
> Gorbunov, E., Tupitsa, N., Choudhury, S., Aliev, A., Richtárik, P., Horváth, S., Takáč, M. Methods for convex $(L_0, L_1)$-smooth optimization: Clipping, acceleration, and adaptivity. arXiv preprint arXiv:2409.14989, 2024
>
> Vankov, D., Rodomanov, A., Nedich, A., Sankar, L., and Stich, S. U. Optimizing $(L_0, L_1)$-smooth functions by gradient methods. arXiv preprint arXiv:2410.10800, 2024.
>
> Laude, E. and Patrinos, P. Anisotropic proximal gradient. arXiv preprint arXiv:2210.15531, 2022.
>
> Maddison, C. J., Paulin, D., Teh, Y. W., and Doucet, A. Dual space preconditioning for gradient descent. SIAM Journal on Optimization, 31(1):991–1016, 2021.

---

> > ### Comment · Reviewer_ZfYL · 2025-04-04
> >
> > Thank you for your responses and clarifications.
> >
> > Typos in the quoted texts were mine. As I have mentioned, my concern was the vague and unclear statements.
> >
> > The authors correctly understood my point from my questions. The responses are sound.
> >
> > Conditioned on fixing the vague statements and specifically clarifying the rasied points on Adam, Adagrad, etc., I think the community would benefit from this work. I raise my score.

---

> > > ### Author Response · Authors · 2025-04-07
> > >
> > > We thank the reviewer for their positive feedback.

---

### Official Review · Reviewer_riQT · 2025-02-26

**Overall Recommendation:** 3

**Summary:**

The paper studies non-linearly preconditioned gradient methods, analyzing algorithms of the form $x^{k+1} = x^k - \gamma\nabla\phi^*(\lambda \nabla f(x^k))$, where where $\phi^*$ is a convex dual reference function, and $f$ is continuously differentiable but potentially non-convex.
This general framework encompasses several widely used methods, including standard gradient descent, AdaGrad without memory (i.e., $\beta_1 = \beta_2 = 0$), Adam with exponential decay rates set to $0$ (i.e., $\beta_1 = \beta_2 = 0$) and clipped GD.
The paper provides theoretical guarantees under a new $(L, \bar{L})$-anisotropic smoothness assumption, extending prior work on generalized smoothness.

**Claims And Evidence:**

The claims are well-supported by theoretical analysis. The proofs appear sound, and the results align with existing literature.

**Essential References Not Discussed:**

I do not see any major omissions.

**Experimental Designs Or Analyses:**

The experiments are designed to demonstrate key theoretical insights rather than provide extensive empirical validation. They are appropriate for this purpose, and the results appear consistent with the claims made in the paper.

**Methods And Evaluation Criteria:**

The paper provides simple experiments for minimizing $\frac{1}{4} \|x\|^4$ and a non-convex phase retrieval task on synthetic data. While limited in scope, these experiments highlight the properties of the proposed framework. Given the paper’s theoretical focus, this level of empirical validation is reasonable.

**Other Comments Or Suggestions:**

No additional comments.

**Other Strengths And Weaknesses:**

**Strengths:**

- The paper is well-written, clearly structured, and engaging.
- It presents an elegant theoretical framework that unifies and generalizes several known algorithms.

**Weaknesses:**

- The significance of the extension is not entirely clear. While mathematically interesting, it would be helpful to better motivate why this generalization is necessary or practically impactful.

**Questions For Authors:**

1. What exactly does the new framework cover that is not already addressed in [2]? A clearer distinction would help contextualize the novelty and significance o contributions.

2. $(L, \bar{L})$-anisotropic smoothness condition reduces to the one in [2] when ${\rm dom} \phi = \mathbb{R}^n$. What additional benefits does this relaxation provide when ${\rm dom} \phi \neq \mathbb{R}^n$?

3. Proposition 2.3 shows that for strongly convex reference functions, the class of anisotropically smooth functions is at least as large as that of Lipschitz smooth ones. What are the most relevant examples demonstrating that this class is strictly larger?

**Relation To Broader Scientific Literature:**

This work builds on prior studies of dual-space preconditioning and anisotropic smoothness.
The general framework considered in the paper was originally analyzed in [1] under the dual relative smoothness condition for convex and essentially smooth functions, and then analyzed in the general non-convex and composite non-smooth setting in [2] under the anisotropic descent property.
The authors extend the previous results by introducing the $(L, \bar{L})$-anisotropic smoothness assumption. This new condition broadens the class of functions considered while reducing to previous formulation when ${\rm dom} \phi = \mathbb{R}^n$.

There is a significant body of literature on generalized smoothness, and this work extends some of these notions.

[1] Maddison, C. J., Paulin, D., Teh, Y. W., and Doucet, A. Dual space preconditioning for gradient descent. SIAM Journal on Optimization, 31(1):991–1016, 2021.

[2] Laude, E. and Patrinos, P. Anisotropic proximal gradient. arXiv preprint arXiv:2210.15531, 2022.

**Theoretical Claims:**

I reviewed the main theoretical results, including key theorems and lemmas. However, I did not verify all details in the supplementary material (additional examples etc).

---

> ### Author Rebuttal · Authors · 2025-03-31
>
> We thank the reviewer for their feedback.
>
> Our answers for **Questions 1 and 2** are summarized in the following. Regarding the $(L, \bar L)$-anisotropic smoothness condition, our contribution is threefold:
> - First and foremost, we describe a general setting where the second-order condition presented in Definition 2.4 is not only necessary but also sufficient for anisotropic smoothness. This condition was shown to be sufficient in (Laude & Patrinos, 2022) only when $f$ itself is a Legendre function, not applying to the general nonconvex setting we consider in our paper. Moreover, in Definition 2.4 we allow for $\phi^*$ to be just Lipschitz smooth and not twice continuously differentiable, thus allowing us to study the interesting case where the update in Equation (2) takes the form of the popular gradient clipping method.
> - Second, we lift a major restriction of (Laude & Patrinos, 2022) by allowing ${\rm dom} \phi \neq \mathbb{R}^n$, which leads to a more general descent inequality by choosing a reference function $\phi$ with faster growth. As an example, through Corollary 2.7 and Proposition 2.9, we get that functions $f \in \mathcal{C}^2(\mathbb{R}^n)$ that are $(L_0, L_1)$-smooth are also $(L, \bar L)$-anisotropically smooth relative to $\phi(x) = -\|x\| - \ln(1-\|x\|)$ which is defined for $\|x\| < 1$. Therefore we obtain novel characterizations for this class of functions (through Proposition 2.2 and Proposition 3.5) and we can tackle problems involving such functions with preconditioned gradient steps defined as in Corollary 2.7.
> - Third, we introduce flexibility by allowing for two different constants whose interplay might have a significant impact in practice, as also indicated by our preliminary experiments.
>
> As for the convergence results, we provide the following novel findings:
> - In the nonconvex setting, our analysis leads to stepsizes up to $2/L$.
> - In the convex setting, we obtain the first sublinear convergence rate for the suboptimality gap $f(x^k)-f^\star$ for large classes of reference functions $\phi$. To the best of our knowledge, such a result does not exist in (Laude & Patrinos, 2022) or in the related literature. We would like to stress that this is not an easy task: in the isotropic case we utilize the star cocoercivity-like property described in Proposition 3.5 and study the problem as a rescaled gradient descent method. In the more challenging separable case we manage to obtain such a result by considering a nonlinear proximal point reformulation of the method and harnessing the subhomogeneity of the reference function.
>
> For **Question 3**: Except for the toy example $f(x) = 1/p \|x\|^p$ considered in the paper, through the aforementioned relation between anisotropic and $(L_0, L_1)$-smoothness, examples of not globally Lipschitz smooth functions that are anisotropically smooth can be found in the $(L_0, L_1)$-smoothness literature (Zhang et al., 2019; Chen et al., 2023). Moreover, we would like to emphasize that examples of Lipschitz smooth functions that are anisotropically smooth with a different parametrization such as the logistic loss function studied in the paper are also of interest, since this can lead to faster algorithms.
>
> ---
> References:
>
> Laude, E. and Patrinos, P. Anisotropic proximal gradient. arXiv preprint arXiv:2210.15531, 2022.
>
> Zhang, J., He, T., Sra, S., and Jadbabaie, A. Why gradient clipping accelerates training: A theoretical justification for adaptivity. arXiv preprint arXiv:1905.11881, 2019.
>
> Chen, Z., Zhou, Y., Liang, Y., and Lu, Z. Generalized-smooth nonconvex optimization is as efficient as smooth nonconvex optimization. In International Conference on Machine Learning, pp. 5396–5427. PMLR, 2023.

---

### Official Review · Reviewer_gkpC · 2025-03-13

**Overall Recommendation:** 3

**Summary:**

1. This paper introduces a unified preconditioning gradient descent scheme for some of the popular optimization algorithms including Adam, Adagrad, gradient clipping.
2. The paper introduces the concept of anisotropic smoothness, which generalizes the relative smoothness. It gives 2nd order sufficient conditions of the concept.
3. The paper gives proof of sub-linear convergence for several classes of reference functions, and verifies through numerical experiments.

**Claims And Evidence:**

1. The suggested scheme is claimed to be a common framework for the popular algorithms. However, it recovers only special cases of parameter choices in those algorithms, and does not provide a framework to analyze them in general.
2. Section 2 supports the claim that the new anisotropic smoothness covers a broader range of functions. In particular, Corollary 2.7 gives a new method of obtaining the forward iterates.
3. Theorem 3.6 and 3.7 gives evidence of sub-linear convergence for nonlinearly preconditioned methods with (1) isotropic reference functions and (2) 2-subhomogeneous functions. However, the claim on general separable reference functions is not explicitly proved and justified.
4. In the numerical experiments, the algorithm is only compared with other basic ones, such as GD and clipping, and results of other preconditioned gradient methods or Adam/Adagrad remain unreported.

**Essential References Not Discussed:**

Relative smoothness is used a lot in the mirror descent method.

Amir Beck, Marc Teboulle, Mirror descent and nonlinear projected subgradient methods for convex optimization, Operations Research Letters, Volume 31, Issue 3, 2003, Pages 167-175.

Filip Hanzely, Peter Richt´arik, and Lin Xiao. Accelerated bregman proximal gradient methods for relatively smooth convex optimization. Computational Optimization and Applications, 79:405 – 440, 2018.

**Experimental Designs Or Analyses:**

The first experiment optimizes a norm-to-power-4 problem, and plots the results of algorithms with different reference functions (but all belong to the class of isotropic functions) and different parameters, though it might be better to give a comparison against other existing algorithms in smooth convex optimization.

The objective of the second experiment has overlap with the first one, as they share the same polynomial growth rate. It can be more convincing to choose other problems. Moreover, the results of the second experiment involves a specific choice of parameters, but in general parameters are unknown, and it might be important to show sensitivity to parameters, like the first experiment.

**Methods And Evaluation Criteria:**

The methods and theoretical proofs make sense, but the paper does not give realistic choices of parameters lambda, gamma for arbitrary objectives. This might be of trouble when the curvature of function is unknown or expensive to compute.

**Other Comments Or Suggestions:**

**Abstract:**

- "thus allowing us to go beyond already established methods." → "thus allowing us to extend beyond already established methods." ("go beyond" is slightly informal)
- "in both the convex and nonconvex setting" → "in both the convex and nonconvex settings" (plural for consistency).

**Other Strengths And Weaknesses:**

Strengths :

1. Writing is clear, the main idea is easy to understand
2. Experiment settings are detailed and easy to implement

Weaknesses:

1. The proposed algorithm in the paper only solves smooth convex optimization problems, and did not touch wider area in optimization.
2. The experimented objective functions are limited to small-scale smooth convex functions defined on the whole space. Non-convex problems and problems on a subset are not experimented.
3. Parameter tuning are not discussed.
4. The intution of some technical results is missing.

**Questions For Authors:**

1. The iteration in (2) and the analysis using relative smoothness resemble the mirror descent update:
   $$
   \nabla \phi(x^{k+1}) - \nabla \phi(x^{k}) = -\alpha_k \nabla f(x_k).
   $$
   What is the advantage of using nonlinear preconditioned gradient methods over mirror descent in this context?

2. Accelerated mirror descent methods have been designed and analyzed. Can the proposed nonlinear preconditioned gradient methods be accelerated in a similar manner?

3. In Theorem 3.6, it is claimed that the norm of the gradient of $f$ decreases monotonically along the iterates of the algorithm:
   $$
   \|\nabla f(x_{k+1})\| \leq \|\nabla f(x_k)\|, \quad \text{for all } k \in \mathbb{N}_0.
   $$
   However, in the simplest case—when the reference function is the squared norm—the method reduces to the standard gradient descent method, for which no such monotonicity result holds. Could you clarify this discrepancy?

**Relation To Broader Scientific Literature:**

The research is closely related to the existing study of non-linear preconditioned gradient descent/mirror descent.

**Theoretical Claims:**

I checked the proofs of Theorem 3.2, Corollary 3.3, and Proposition 3.5. No issues were found in these proofs.

---

> ### Author Rebuttal · Authors · 2025-03-31
>
> We appreciate the reviewer’s time and effort in providing feedback.
>
> Regarding the stated **Weaknesses**:
> 1. Our results cover both the convex and nonconvex setting. The convergence results in the nonconvex setting are presented in Section 3.1.
> 2. The scope of the current manuscript is to lay the theoretical foundations of the framework. Therefore, we have provided indicative experiments on problems that are relevant in the generalized smoothness literature. Moreover, we note that the phase retrieval problem is nonconvex.
> 3. As is standard for descent-type methods we provide a sufficient second order condition that allows us to compute the stepsizes of the method. In cases where the constants are difficult to compute, one can utilize a variation of the classical linesearch procedure to adaptively determine the stepsize sequence.
> 4. We tend to disagree with the reviewer as we believe that looking at these algorithms under the lens of generalized convexity provides valuable intuition. This connection with generalized convexity is also the main ingredient for many of our proofs.
>
> Concerning item 3 of the **Claims And Evidence** section we would like to stress that our claim is that we prove with a novel technique the sublinear convergence rate for large classes of functions, a result which is presented in Theorem 3.7. Regarding item 1 of the **Claims And Evidence** section, see our response to Reviewer ZfYL.
>
> Answer to **Question 1**:
>
> Indeed, the update defined in (2) bears similarities to mirror descent. In fact the scheme we analyze can be considered as left-preconditioning in contrast to mirror descent’s right-preconditioning (Maddison et al., 2021, p. 3). Nevertheless, the two methods produce different iterates while the descent inequalities used to analyze them also differ. Our paper relies on anisotropic smoothness, while mirror descent is analyzed using the relative smoothness condition. Thus, the nonlinearly preconditioned gradient scheme is used to tackle different problem classes than mirror descent.
>
> Answer to **Question 2**:
>
> To the best of our knowledge, Bregman gradient descent schemes under relative smoothness can be accelerated when the generated Bregman divergence satisfies some favorable properties (Hanzely et al., 2021). We believe that this is the case with the nonlinearly preconditioned gradient scheme as well, i.e. the method can be accelerated under specific conditions for the reference function $\phi$. We believe that this is interesting future work.
>
> Answer to **Question 3**:
>
> We respectfully disagree with the reviewer. It is known that for convex and Lipschitz smooth functions, gradient descent with a fixed stepsize of $1/L$ monotonically decreases the norm of the gradient (Beck, 2017, Theorem 10.27). We thus believe that our Theorem 3.6 generalizes this result to the anisotropic smoothness framework for isotropic reference functions.
>
> ---
>
> References:
>
> Maddison, C. J., Paulin, D., Teh, Y. W., and Doucet, A. Dual space preconditioning for gradient descent. SIAM Journal on Optimization, 31(1):991–1016, 2021.
>
> Hanzely, F., Richtarik, P., & Xiao, L. (2021). Accelerated Bregman proximal gradient methods for relatively smooth convex optimization. Computational Optimization and Applications, 79, 405-440.
>
> Beck, A. First-order methods in optimization. SIAM, 2017.

---

### Official Review · Reviewer_MxEp · 2025-03-14

**Overall Recommendation:** 5

**Summary:**

The studies a non-linearly preconditioned gradient descent method for optimizing some scalar functions $f:R^n\to R$.
The iterations of the method are of
$$x_{k+1}=x^k-\gamma \nabla \psi^*(\lambda \nabla f(x^k)),$$
which is similar to the approach of "Dual Space Preconditioning for Gradient Descent" by Maddison et al, although that paper did not consider the inclusion of the parameter $\lambda$.

The original paper Maddison et al. only shown convergence results for convex functions $f$, and convex reference functions $\psi$.
Recent works Laude & Patrinos, 2022 and Laude et al., 2023 have shown convergence to stationary points for non-convex functions $f$ under a condition called the anisotropic descent property.

The key contribution of this paper is the development of a very general $(L,\overline{L})$-anisotropic smoothness property, which is shown to be easily verifiable for a broad range of non-convex functions in Lemma 2.5.
Under this property, the algorithm is shown to converge to stationary points of $f$ with explicit rates. Additional sharper convergence results are also obtained for convex $f$ in terms of the optimality gap $f(x^k)-f(x^*)$, improving upon earlier results in Maddison et al.

Some numerical results are also presented to show the practical performance of the algorithm compared to gradient clipping.

**Claims And Evidence:**

This is a theory paper. The claims of the paper are well supported by the mathematical proofs. The numerical illustrations are of secondary importance, but they also show good empirical performance.

**Essential References Not Discussed:**

I believe that all essential references have been discussed.

**Experimental Designs Or Analyses:**

The numerical experiments in the paper are appropriately designed, and the analysis of the numerical results is appropriate.

**Methods And Evaluation Criteria:**

The proposed methods and evaluation criteria seem reasonable to us. The method was evaluated both on a convex as well as a non-convex test problem, and compared to gradient clipping.

**Other Comments Or Suggestions:**

No other comments.

**Other Strengths And Weaknesses:**

The key contribution of this paper is the development of a very general $(L,\overline{L})$-anisotropic smoothness property, which is shown to be easily verifiable for a broad range of non-convex functions in Lemma 2.5.

Under this property, the algorithm is shown to converge to stationary points of $f$ with explicit rates. Additional sharper convergence results are also obtained for convex $f$ in terms of the optimality gap $f(x^k)-f(x^*)$, improving upon earlier results in Maddison et al.

These results are interesting as the assumptions on the function $f$ are very mild, the algorithm is simple, and the results are fully explicit.

One weakness of the paper is that the algorithm presented is not new, and it has been studied before in the general non convex and composite non-smooth setting in Laude & Patrinos, 2022 different conditions.
Nevertheless, the theoretical results obtained in this paper are shown under much simpler conditions and the results are stronger.

**Questions For Authors:**

Could you please further clarify the differences between the conditions in your current paper, and the conditions used in Laude & Patrinos, 2022?

In terms of numerical experiments, in the context of non-convex functions, it would be interesting to see a neural network example for a small dataset using full gradients (with sufficiently wide networks, the loss can be close to 0 on the training data). In particular, you could do a comparison with gradient clipping in such scenarios. Do you think that the performance advantages would be similar as in Section 4.2?

**Relation To Broader Scientific Literature:**

This algorithm is closely related the algorithm presented in the paper "Dual-space preconditioning for gradient descent", as well as papers by Lu et al. and Bauschke et al.
A variant of present algorithm has been studied before in the general non convex and composite non-smooth setting in Laude & Patrinos, 2022 different conditions.
Nevertheless, the theoretical results obtained in this paper are shown under much simpler conditions and the results are stronger.
The authors do a good job at citing relevant papers from the literature.

**Theoretical Claims:**

I have looked at the proofs, and checked the correctness of the main results (Theorems 3.2, 3.6 and 3.7). I believe that the presentation is clearly written and the proofs are correct and rather ingenious.

---

> ### Author Rebuttal · Authors · 2025-03-31
>
> We thank the reviewer for their positive feedback.
>
> Answer to **Question 1**:
>
> There are three major differences in the conditions considered in the current paper compared to (Laude & Patrinos, 2022):
> 1. We consider reference functions $\phi$ of possibly not full domain. This leads to a more general descent inequality that allows us to tackle larger problem classes. As an example of the significance of this extension, through Corollary 2.7 and Proposition 2.9 we obtain that functions $f \in \mathcal{C}^2(\mathbb{R}^n)$ that are $(L_0, L_1)$-smooth are also $(L, \bar L)$-anisotropically smooth relative to $\phi(x) = -\|x\| - \ln(1-\|x\|)$, a function which is defined for $\|x\| < 1$. We moreover focus on strongly convex reference functions in order to capture a function class that is wider than Lipschitz smoothness through Proposition 2.3.
>
>  2. We provide a sufficient second order condition for anisotropic smoothness in the form of Definition 2.4, combined with Proposition 2.9. Such a condition (as described in Lemma 2.5) was shown in (Laude & Patrinos, 2022) to be in general only necessary, while it was shown to be sufficient under Legendre convexity of $f$ in (Laude & Patrinos, 2022, Proposition 4.1). Thus we overcome a major limitation of (Laude, Patrinos, 2022) in that we provide a computable condition for anisotropic smoothness. Moreover, in Definition 2.4, we relax the continuous differentiability of $\nabla \phi^*$ to Lipschitz continuity thus allowing us to cover more interesting algorithms. Central to our analysis is the new result regarding the (strong) monotonicity of the forward operator.
>
> 3. In our paper we study the convex case separately, obtaining for the first time sublinear convergence rates for large classes of reference functions of both the isotropic and the separable type.
>
> Answer to **Question 2**:
>
> We thank the reviewer for their suggestion. We performed a preliminary experiment for a subset of the MNIST dataset (600 data points) on a four-layer fully connected network with layer dimensions [28 × 28, 128, 64, 32, 32, 10] and ReLU activation functions, using the cross-entropy loss function. We compare the nonlinearly preconditioned gradient method generated by $\phi_1(x) = -\|x\|-\ln(1-\|x\|)$ and $\phi_2(x) = \cosh(\|x\|)-1$ and the gradient clipping method. We tuned the methods by performing a grid search on the hyperparameters. The results are presented in the following table, where we report the training loss every 30 iterations.
>
> | Method   | k=30     | k=60     | k=90     | k=120    | k=150    | k=180    | k=210    |
> |----------|----------|----------|----------|----------|----------|----------|----------|
> | $\phi_1$ | 0.975662 | 0.440863 | 0.118921 | 0.096205 | 0.000098 | 0.000035 | 0.000020 |
> | clipping | 0.921261 | 0.277707 | 0.130072 | 0.145052 | 0.068124 | 0.054605 | 0.000012 |
> | $\phi_2$ | 1.308716 | 0.593909 | 0.300052 | 0.101221 | 0.008348 | 0.001282 | 0.000670 |
>
> In this experiment it seems that the method generated by $\phi_1$ is the best performing, while the clipping gradient method speeds up towards the end to achieve better final accuracy. Due to the simplicity of the experiment we cannot conclude on the effectiveness of the compared methods in general. We believe that different reference functions lead to performance advantages on different architectures and loss functions.
>
> ---
>
> References:
>
> Laude, E. and Patrinos, P. Anisotropic proximal gradient. arXiv preprint arXiv:2210.15531, 2022.

---

> > ### Comment · Reviewer_MxEp · 2025-04-04
> >
> > Thank you for answering my questions during the rebuttal. I feel that this is a nice contribution, with solid theoretical results and the rebuttal also obtained so interesting preliminary results for neural networks. The authors could include more detailed studies on such problems in the final version. I increase my score.

---

> > > ### Author Response · Authors · 2025-04-07
> > >
> > > We thank the reviewer for their positive evaluation of our work. If the paper gets accepted, we will include more detailed results on our experiments.

---

### Decision · Program_Chairs · 2025-05-01

**Decision:**

Accept (oral)

**Comment:**

The paper presents a unified nonlinear gradient preconditioning framework that encompasses popular algorithms such as gradient clipping, Adam, Adagrad, and recent methods tailored for $(L_0, L_1)$-smooth functions. It also introduces anisotropic smoothness that is more general that L-smoothness, and showcases its application in the convergence analysis of the algorithm in the nonconvex regime.

Four reviewers evaluated the paper, and their overall assessment is positive. I agree with their evaluation and believe the paper makes a significant contribution with compelling results. In particular, I believe the most impressive result in this paper is the introduction of the anisotropic smoothness property, which is shown to be easily verifiable for a broad range of non-convex functions.